



# Holocene vegetation transitions and their climatic drivers in MPI-ESM1.2

Anne Dallmeyer[1], Martin Claussen[1,2], Stephan J. Lorenz[1], Michael Sigl[3], Matthew Toohey[4], Ulrike Herzschuh[5,6,7]

[1]Max Planck Institute for Meteorology, Bundesstrasse 53, 20146 Hamburg, Germany
[2]Centrum für Erdsystemforschung und Nachhaltigkeit (CEN), Universität Hamburg, Bundesstrasse 55, 20146 Hamburg, Germany
[3]Climate and Environmental Physics and Oeschger Centre for Climate Change Research, University of Bern, Bern, Switzerland
[4] Department of Physics and Engineering Physics, University of Saskatchewan, Saskatoon, Canada
[5]Alfred Wegner Institute, Helmholtz Centre for Polar and Marine Research, Potsdam, Germany.
[6]Institute of Environmental Sciences and Geography, University of Potsdam, Germany
[7]Institute of Biochemistry and Biology, University of Potsdam, Germany

*Correspondence to*: Anne Dallmeyer (anne.dallmeyer@mpimet.mpg.de)

**Abstract**

We present a transient simulation of global vegetation and climate patterns of the mid and late Holocene using the MPI-ESM (Max Planck Institute for Meteorology Earth System Model) at T63 resolution. The simulated vegetation trend is discussed in the context of the simulated Holocene climate change. Our model captures the main trends found in reconstructions. Most prominent are the southward retreat of the northern treeline that is combined with the strong decrease of forest in the high northern latitudes during the Holocene and the vast increase of the Saharan desert, embedded in a general decrease in precipitation and vegetation in the northern hemispheric monsoon margin regions. The southern hemisphere experiences weaker changes in total vegetation cover during the last 8000 years. However, the monsoon-related increase in precipitation and the insolation-induced cooling of the winter climate lead to shifts in the vegetation composition, mainly between the woody plant functional types (PFTs).

The large-scale global patterns of vegetation almost linearly follow the subtle, approximately linear, orbital forcing. In some regions, however, non-linear, more rapid changes in vegetation are found in the simulation. The most striking region is the Sahel-Sahara domain with rapid vegetation transitions to a rather desertic state, despite a gradual insolation forcing. Rapid shifts in the simulated vegetation also occur in the high northern latitudes, in South Asia and in the monsoon margins of the southern hemisphere. These rapid changes are mainly triggered by changes in the winter temperatures, which go into, or move out of, the bioclimatic tolerance range of individual PFTs (Plant Functional Types). The dynamics of the transitions are determined by dynamics of the Net Primary Production (NPP) and the competition between PFTs. These changes mainly occur on timescales of centuries. More rapid changes in PFTs that occur within a few decades are mainly associated with the time scales of mortality and the bioclimatic thresholds implicit in the dynamic vegetation model, which have to be interpreted with caution.

Most of the simulated Holocene vegetation changes outside the high northern latitudes are associated with modifications in the intensity of the global summer monsoon dynamics that also affect the circulation in the



extra tropics via teleconnections. Based on our simulations, we thus identify the global monsoons as the key

player in the Holocene climate and vegetation change.

## 1 Introduction

The Holocene (the last ~11,500 years) has been a period of strong global environmental changes. These were mainly forced by the retreat of the continental ice masses and variations in the seasonal insolation associated with changes in the Earth orbital constellation and solar variability (Mayewski et al., 2004).

During the early-Holocene, the earth approached perihelion (the point closest to the sun on the earth orbit) during August (9ka BP = 9000 years before present). Due to the precession of the equinoxes and the earth

axis, the timing of perihelion shifts continuously, roughly by one month every 1900 years (Ruddiman, 2008). This leads to gradual changes in the seasonal energy input from the sun. For instance, the northern hemisphere received approx. 10% more summer insolation at 9ka, whereas winter insolation was reduced by approx. 10%, leading to only small changes in the annual mean (Berger, 1978) but a higher amplitude of the seasonal cycle.

The climatic response is regionally amplified by feedbacks in the climate system, transforming the seasonal forcing into a response in the annual mean signal (e.g. Crucifix et al., 2002). Winter climate was colder and summer climate was warmer on the northern hemispheric continents during the mid-Holocene (e.g. Zhang et al., 2018, Viau et al., 2006, Davis et al., 2003).

Fewer studies have focused on the temperature evolution on the southern hemisphere. Model experiments

reveal a cooling trend in austral winter and a warming trend in austral summer during the Holocene (Lorenz et al., 2006). Hence, the seasonality in the northern hemisphere weakens, while it strengthens in the southern hemisphere (e.g. Fischer and Jungclaus, 2011). As a consequence, the atmospheric and oceanic circulations continuously adjust to the energy change, affecting in turn the precipitation and temperature distribution and thus also the global vegetation patterns.

The decrease in boreal summer insolation induced a southward movement of the Intertropical Convergence Zone (ITCZ, e.g., Haug et al., 2001) and a southward retreat and weakening of the summer monsoons in the northern hemisphere (e.g. Kutzbach 1981, Liu et al. 2003, Dallmeyer et al., 2015, D'Agostino et al., 2019), implying a drying trend in subtropical Asia, Africa and Central America. In contrast, the summer monsoon systems in the southern hemisphere strengthen during the Holocene (e.g. Zhao and Harrison, 2012, Jiang et

al,. 2015, Prado et al. 2013a) and result in a moistening trend roughly north of 40°S (e.g. Royas and Moreno, 2009).

Reconstructions furthermore reveal warmer and drier mid-Holocene conditions in large parts of North America (Viau and Gajewski, 2001, Williams et al., 2010) and South America (Lamy et al, 2001, Prado et al., 2013b). As possible explanations for this in-phase climate change in the Americas, Holocene weakening

of the northern hemispheric subtropical anticyclones and an enhancement of the north-southward migration





of the ITCZ have been proposed (Grimm et al., 2001). Furthermore, the southern hemispheric westerly winds were probably shifted poleward and were intensified during the mid-Holocene, but reconstructions reveal contradictory signals (Fletcher and Moreno and references therein, 2012).

The global climate was also affected by reduced El Niño Southern Oscillation activity and a more La Niña like mean state of the Tropical Pacific during the mid-Holocene (e.g. Clement et al., 2004; Overpeck and Webb, 2000), leading particularly to wetter conditions in Australia and drier conditions on the extratropical American continents (Barr et al., 2019). A detailed summary of the mid- to late-Holocene climate change is given in Wanner et al. (2008).

Pollen-based reconstructions reveal substantial vegetation changes as response to the Holocene climate signal. So far, most studies deal with the vegetation dynamics at individual sites. Several continental-scale syntheses are available (Cao et al., 2019a), but no global overview exists, showing the Holocene vegetation trend. This is also because spatial coverage of records is low outside Europe and North America. Even more systematic investigations of vegetation-climate relationships are not possible because climate proxy-data independent from vegetation change are scarce.

Pollen records indicate an expansion of forests in the northern mid- to high-latitudes, replacing tundra areas, (MacDonald et al., 2000, Prentice et al., 2000) and reveal a northward shift of the northern treeline during mid-Holocene compared to pre-industrial distributions (Bigelow et al., 2003).

In the Afro-Asian monsoon region, the vegetation belts penetrated further inland and the northern hemispheric desert region were substantially reduced during mid-Holocene (Zhao et al., 2009; Feng et al., 2006; Yu et al., 2000; Jolly et al., 1998, Metcalfe et al, 2015). At least for North Africa, reconstructions suggest a rapid increase of the Saharan desert during the Holocene (e.g., deMenocal et al., 2000; Shanahan et al. 2015).

The vegetation change in North America is complex, as it occurs rather on taxa level. Generally, plant populations expand northwards during the Holocene (Williams et al., 2004). Biome reconstructions show rather stable conditions, but also reveal a northward expansion of the temperate forest and evergreen taiga (Cao et al., 2019a). Furthermore, forest taxa re-invade into the prairie, reflecting the increase in precipitation during the Holocene (Grimm et al., 2001). Most of the American savanna regions were more widespread during the mid-Holocene (Behling and Hooghiemstra, 2000).

In the Amazonian region, vegetation changes were minor, with the exception of southeast Brazil and southwest Amazonia indicating an increase in forests during the Holocene. Southern hemispheric grasslands reveal a transition to communities requiring more moisture during the Holocene reflecting the increase in moisture levels (Grimm et al., 2001).

Biome reconstructions for Australia show only little and locally varying vegetation changes since the mid-Holocene. Some records from the south-eastern part reveal moisture-stressed vegetation types at 6ka BP (i.e. before present), implying drier conditions there. Other records, e.g. from the Great Diving Range indicate increased vegetation cover and more moisture demanding plants during the mid-Holocene. The biome changes in the tropics are minor, rather showing transitions between different forest types (Pickett el al., 2004).





Reconstructions for Southern Africa are rare, often incomplete, and show contradictory signals. During the
early Holocene, desert pollen concentrations in sediments off the western South African coast were
increased, followed by a period of increasing moisture during the mid-Holocene. Vegetation was more open
on the southern continent (Scott and Lee-Thorp, 2004). Numerous records indicate an aridification during the
mid- and late-Holocene (Burrough and Thomas, 2013). In tropical south east Africa, more humid vegetation
types occurred during the mid-Holocene (e.g. Marchant et al., 2018).

These assessments indicate that we still lack substantial understanding of vegetation dynamics in response to
climate change not only at the global scale, but also at the regional to continental scale.

Global simulations of Holocene vegetation change hitherto mostly focused on the comparison of vegetation
composition between time-slices, while the dynamics were seldom taken into consideration (e.g. Brovkin et
al., 2002; Crucifix et al., 2002). Recently, Braconnot et al. (2019) discussed the advantages and challenges of
transient vegetation simulations in comparison to time-slice simulations by using the Earth System Model
IPSL. In their simulation, the vegetation reduction in Northern Africa occurs and ends later than the forest
decline in the northern hemisphere. In addition, the model indicates large variations in the forest composition
throughout the Holocene. However, the vegetation-climate relationships in time and space were hitherto not
investigated systematically in transient vegetation simulations conducted with Earth system models.

Here, we use a transient simulation of the mid- to late Holocene period performed with the comprehensive
Earth System Model MPI-ESM1.2 (Bader et al., 2020) to analyse the Holocene vegetation change in the
context of the simulated global climate change. We focus on the long-term, multi-centennial vegetation
trends. As some reconstructions reveal abrupt transitions in the vegetation composition in some regions over
the Holocene, we test the simulated vegetation changes with respect to their rapidity compared to the long-
term change in the model.

After the description of the model and the transient simulation, the simulated vegetation change is evaluated
against global biome-reconstructions for the mid-Holocene and present-day time-slices.

In section 4 we provide an overview of the main global vegetation trends and assess the climatic drivers by a
redundancy analysis. Afterwards, the vegetation trends are discussed on the regional level and interpreted in
the context of the changes in the climate and large-scale atmospheric circulation systems (section 5). The
simulated rapid vegetation changes during the Holocene are discussed in section 6. We summarize the results
and conclude our study in section 7.

## 2 Methods

### 2.1 The Earth System Model MPI-ESM1.2

The model MPI-ESM1.2 (Mauritsen et al., 2019) developed at the Max-Planck-Institute for Meteorology
consists of the general circulation model of the ocean MPIOM (Jungclaus et al., 2013), including the ocean
biogeochemistry model HAMOCC (Ilyina et al., 2013), coupled to the atmospheric general circulation model
ECHAM6.3 (Stevens et al., 2013). The terrestrial carbon cycle and vegetation dynamics are calculated by the
land-surface scheme JSBACH3 (Reick et al., 2013) that incorporates the dynamic vegetation module





developed by Brovkin et al. (2009). Through the inclusion of the subsystem models JSBACH and
      HAMOCC, the full carbon cycle is considered in MPI-ESM. Nevertheless, the carbon cycle is not fully
      interactive in this simulation as the atmospheric $CO_2$ concentration is prescribed. ECHAM6 and the land
      model was configured with a spectral resolution of T63 (corresponding to a horizontal resolution of
      approximately 200km on a Gaussian grid) with 47 levels in the vertical atmosphere. For the ocean model the
horizontal resolution GR15 (i.e. 256x220 on a bipolar grid, corresponding to 12 to 180km) and 64 vertical
      levels have been used.

### 2.2 The dynamic vegetation module

      In the dynamic vegetation module (see Brovkin et al., 2009, Reick et al., 2013), natural vegetation is
represented by eight different plant functional types (PFT). Trees are classified into either tropical or
      extratropical trees and their phenology can be evergreen or deciduous. Shrubs are distinguished as raingreen
      (moisture limited) and cold-resistant (temperature limited) shrubs. Grass is differentiated between C3 and C4
      grass. JSBACH uses a tiling approach, thus all PFTs can coexist in each grid-cell of the land-surface in
      principle and are assigned certain cover fractions, ranging from 0 (not present) to 1 (covering the full grid-
cell).
      The occurrence of each PFT is limited by temperature constraints that reflect the bioclimatic tolerance of the
      individual PFTs. These environmental thresholds represent the chilling requirements of the plants by the
      maximum mean temperature of the coldest month ($Tc_{max}$) and the cold resistance of the plants by the
      minimum mean temperature of the coldest month ($Tc_{min}$). Furthermore, the presence of the PFTs is regulated
by their heat requirements during their growth phase. This limit is realized by temperature sums over days on
      which mean temperature exceeds different thresholds (0°C and 5°C), called growing degree days (GDD0 and
      GDD5, respectively). Cold shrubs are also not able to survive in too warm climates and are limited by the
      maximum mean temperature of the warmest month ($Tw_{max}$). The different bioclimatic limits for the
      individual PFTs are listed in Table 1.
In addition to these limits, the dynamics of changes in the fractional coverage of PFTs in response to climatic
      change are governed by the dynamics of the Net Primary Production (NPP) of the competing PFTs. The time
      scales of NPP dynamics include time scales for allocation, mortality and perturbation. The times scales of
      allocation and mortality are the same for each individual PFT, but differ between PFTs. They range from 50
      years for extratropical trees to 1 year for grass (see Table 1). If climate changes and if different PFTs can
exist within the range of climate variability, then the PFT with the strongest NPP dominates, and the time
      scale of the transition mainly depends on the difference in the NPP allocation of the competing PFTs. If this
      difference is small, the transition is slow. The transition can be faster, if one of the PFTs has much shorter
      time scales of allocation and mortality. The fastest changes occur, if the climate crosses a bioclimatic
      threshold (see Table 1) or if a large-scale perturbation, such as windbreak or fire, occurs. Then, the PFT
affected decays exponentially within a few decades or less.



The fractional PFT coverage of a grid cell can also change, if the bare soil fraction (BSF), which represents the seasonal and permanently non-vegetated ground, varies. The BSF is determined by the relation of the carbon actually stored in the carbon pool for living tissues via the NPP of the individual plants to the maximum yearly carbon storage in this pool. This maximum is defined by the carbon costs needed by the plants to build up the leafs, stems etc. completely and depends on the ratio of the maximum leaf area index to the specific leaf area of the PFTs.

### 2.3 Transient simulation

To run the model in quasi-equilibrium between mid-Holocene boundary conditions, climate and the carbon cycle, a "spin-down" simulation has been performed with all forcings fixed to the values of the year 6000 BCE. This simulation was initialised from the pre-industrial climate and vegetation state and was run for more than 1000 years. The transient simulation then started from this equilibrium state and continued until pre-industrial times (i.e. 1850 CE). The transient simulation used in this study has been performed nearly identical to the simulation described in Bader et al. (2020), Brovkin et al. (2019), and Dallmeyer et al. (2020). The following forcings were prescribed:

a) orbital-induced insolation changes (Berger, 1978), updated every decade. These mainly impact the seasonal cycle of e.g. temperature and precipitation. The seasonality increases in the southern hemisphere and reduces in the northern hemisphere during the Holocene.

b) Greenhouse gas concentration (methane, carbon dioxide and nitrous oxide) inferred from ice core records (F. Joos, personal communication; see Brovkin et al., 2019 and Köhler, 2019), updated every decade. The difference in $CO_2$ between start and end of the simulations amounts to approximately 20 ppm.

c) Stratospheric sulfate aerosol injections imitating volcanic eruptions, prescribed from the Easy Volcanic Aerosol (EVA) forcing generator (Toohey et al., 2016), read annually, but calculated daily by linear interpolation. This forcing has been reconstructed based on ice-cores from both hemispheres to better identify volcanic eruptions with global impact on the climate. From Greenland, we employed sulfate data from the Greenland Ice Sheet Project Two (GISP2; 72.97°N, 38.80°W) ice core (Mayewski et al., 1997). From Antarctica, we used sulfate data from the EPICA Dronning Maud Land (EDML; 75.00°S, 00.07°E) and Dome Concordia (EDC; 75.10°S, 123.35°E) ice cores (Severi et al., 2007), as well as sulfur and sulfate data from the WAIS (West Antarctic Ice Sheet) Divide ice core project (WD; 79.48°S, 112.11°W) (Cole-Dai et al., 2021). These four ice cores were synchronized to the WD2014 chronology (Sigl et al., 2016) and volcanic sulfate mass deposition rates over Greenland and Antarctica were estimated using established methods (Sigl et al., 2014; Sigl et al., 2015). Using the methodology developed (Gao et al., 2007) and applied to a similar bipolar network of ice cores over the past 2,500 years (Toohey and Sigl 2017) we estimated stratospheric sulfur injection (SSI) for in total 528 volcanic eruptions. Conversion of SSI into the aerosol properties is performed with the Easy Volcanic Aerosol module (Toohey et al. 2016). Aerosol extinction is assumed to be linearly proportional to mass for eruptions smaller than the 1815 Tambora eruption, and follows a 2/3 power-law scaling for larger eruptions (Crowley and Unterman, 2013). The previous volcanic forcing reconstruction



used in Bader et al. (2020) was based on a Greenland ice-core (GISP2, Zielinski et al. 1996) only and therefore overestimated the effect of Icelandic and other extratropical volcanic eruptions on the global climate.

d) Spectral Solar Irradiance forcing, includes extrapolated 11-years solar cycle based on sun-spot observations data-sets of far infrared, near infrared and visible radiation (Krivova et al., 2011), read annually, but calculated daily by linear interpolation.

e) LUH2 land-use forcing by Hurtt et al. (2020) conform to the CMIP6 simulations, read annually, but calculated daily by linear interpolation. This forcing begins 850 CE with a quasi linear transition period (1000 years) starting 150 BCE to slowly build up the land-use. Bader et al. 2020 used a preliminary (unpublished) version of the LUH2 dataset.

In the analysis of this study, the period 6000BCE to 150BC is considered only, to exclude the effect of land use on the vegetation distribution. For easier nomenclature, we define the mid-Holocene time slice (further referred to 8ka before the year 2000, i.e. b2k) by the climatological mean of the first 100 years of this transient simulations (i.e. year 6000-5901 BCE) and the late-Holocene reference period (further referred to 2.15ka b2k) by the climatological mean of the last 100 years of this simulations without land-use (i.e. year 250-150 BCE). Accordingly, we define the 6ka b2k time-slice by the climatological mean of the years 4000-3901 BCE.

**2.4 Analysis methods**

To summarise the Holocene vegetation change, we choose the fuzzy c-means clustering technique (Bezdek,1981). This method calculates a set of independent cluster centres (here: time-series of vegetation development), such that the sum of the square distances between the items assigned to the cluster and the cluster centre is minimized. This means, that grid-cells showing similar vegetation changes over the course of the Holocene are grouped into the same cluster. We here use the c-means clustering function in R (Meyer et al., 2017) and calculate the cluster on the basis of 100yr. running mean time-series for the PFT-groups "forest", "shrubs", "grass" and "bare soil". The number of independent cluster has been optimized by the Xie-Beni method (Xie and Beni, 1991).

To test the most important climatic driver in the different regions, we performed a constrained ordination via the redundancy analysis function "rda" implemented in the "vegan" package of R (Oksanen et al, 2018).

The redundancy analysis (RDA) is an extended multiple linear regression method and measures the linear relationship between a set of response variables (here the PFT-groups "forest", "grass", and "shrubs") and explanatory variables (here precipitation, temperature of the warmest month and temperature of the coldest month). It ordinates the vegetation on axes that are constructed in such a way that their relationship to linear combinations of the climate variables is maximised. Therefore, the RDA technique can also be seen as a constrained version of the principal component analysis. In this study, it is taken as a measure of the variation in vegetation that can be explained by the climate variables. The temperature of the coldest month explains only a small part of the variance in an initial RDA and was thus excluded in the final RDA shown here.




## 3 Evaluation of the simulated vegetation distribution against biome reconstructions

So far, palaeo-vegetation reconstructions have mostly been synthesized by biome distributions for individual time-slices (e.g. Harrison, 2017; Cao et al., 2019a). Quantitative vegetation reconstructions are only available for few regions (e.g. Trondman et al., 2015; Cao et al., 2019b, Li et al., 2020). Therefore, we decided to

evaluate the simulated vegetation against pollen-based biome reconstructions for the mid-Holocene (6ka). The simulated PFT cover fractions were converted into mega-biome distributions using the approach by Dallmeyer et al., (2019). The agreement was tested by using the Best Neighbour Score (BNS, Tab.2, cf. Dallmeyer et al., 2019) that considers also the agreement in the surrounding grid-cells of the record sites. Fig. 1 shows the resulting simulated (a) and reconstructed biome distribution (b, after Harrison, 2017) for

6ka. In principle, the vegetation in the model is well in line with the reconstructions (BNS=0.73). All major biome belts are reproduced. In particular, the extratropical forest biomes are well represented showing a BNS of 0.79 (boreal forest) and 0.88 (temperate forest). The simulated deserts (BNS=0.39) and savannas (BNS=0.12) show little correspondence with the reconstructions. On the one hand, this is related to the fact that also in this simulation the northward spread of vegetation into the Sahara during the Mid-Holocene is

underestimated compared to the reconstructions (cf. Dallmeyer et al., 2020). On the other hand, the extent of the desert area in Central Asia is too small. The reconstructed desert in the southwest of North America (mainly around the Mojave and Sonoran desert) is not reflected in the model. Very few reconstructions exist for the savanna regions, probably biasing the comparison. However, the method for biomizing PFTs has shortcomings in mapping savannas as this biome exhibits a very distinct ecology and therefore cannot solely

be delimited by climatic constraints (cf. Dallmeyer et al., 2019).

To analyse the main change in biomes towards PI, the biomes were further grouped into the main categories forests, savannas, grassland/tundra, and deserts. Fig.1c-d shows qualitatively the simulated and reconstructed difference for these biome categories between 6ka and PI on the basis of the ecological development. Positive development designates the transition from a more open landscape to a tree-dominated landscape

(e.g. desert in 6ka to grassland, grasslands to savannas or to forests in PI). Similarly, negative development refers to the transition from tree-dominated biomes to a more open landscape (e.g. forest in 6ka to grassland in PI).

Since not all reconstructions mapped for 6ka reach into pre-industrial times, we also considered PI records in the grid-cells surrounding those of sites mapped in 6ka. This method does not lead to a significant

improvement of the record density in the difference plot. Therefore, we retained the original selection, i.e. showing only records with a counterpart at PI. Both, the model and the reconstructions, show only little change in the area covered by these main biomes. Most remarkable is the decrease in forest and grassland in the northern hemispheric monsoon regions and the decrease of forest along the Taiga-Tundra boundary in the high northern latitudes indicated by the model. The reconstructions in principle also suggest this

development, even if the record density is too low to show a detailed picture, particularly in North Africa.



The decrease in openness in central North America and central Asia is visible in the model and in the reconstructions. The records furthermore suggest a greening at the Labrador Sea coast, which is not shown by the model. This may be related to the migration of plants after the retreat of the Laurentine ice-sheet. The changes in land ice coverage are not considered in the model so that this process cannot be represented. The

model indicates a positive ecological development in the South African and South American extratropics. This is not seen in the records showing rather no change in main biome category, but only few records exist that are located directly in the regions in which the model shows this positive signal. Hence our prediction, or hindcast, awaits further evaluation.

**4 Global view of the simulated Holocene vegetation and climate trend**

The simulated Holocene vegetation change from 8ka b2k to 2.15ka b2k has been clustered on the basis of the 100yr-running mean time-series using a c-means clustering technique in R (Meyer et al., 2017). The desert and forest change can be grouped in four, the shrubs and grass change in five different cluster (Fig. 2). Figure 2a,c,e,h mark the regions that can be assigned to the respective cluster. Figure 2b,d,f,g show the trends in the respective cluster centre. Overall, the Holocene vegetation changes in this global view are quite linear,

overlaid by weak multi-centennial variability. However, this does not imply that non-linear changes in PFTs do not occur on a local or even regional level. The most striking vegetation shifts seen in Fig. 2 are:

    a) a strong decrease in forest and increase in grass and shrubs in the high northern latitudes reflecting the southward retreat of the northern treeline and the shift from Taiga forest to Tundra

    b) a (strong) increase in the subtropical northern hemispheric deserts coinciding with a decrease and

equator-ward retreat of all vegetation types in the monsoon regions

    c) an increase in the vegetated area in extratropical North America (mainly 30°-60°N, 90-120°W) related to an increase in forest and partly grass

    d) an increase in the vegetation in extratropical South America and South Africa due to an increase in forest, grass and shrubs (S Africa), including the southern hemispheric monsoon regions

e) a bipolar vegetation change in Australia, mostly driven by an increase in grass on the northern and a decrease of grass on the southern part of the continent.

Fig. 3 shows a similarly calculated cluster analysis for the 100yr running means of annual mean precipitation, and the temperature of the warmest and coldest month, respectively, assuming that these variables are the main climatic factors affecting the vegetation change. In line with the increasing wintertime

insolation during the Holocene, the temperature of the coldest month ($T_{cold}$) uniformly increases in the northern hemisphere, in particular on the continents by up to 1.6K (cluster center). In the southern hemisphere, $T_{cold}$ does only change in a few grid-boxes in South Africa and in the northern part of Australia, showing an overall decreasing trend with strong millennial variability.

The temperature of the warmest month ($T_{warm}$) shows the opposite trend, in line with the summertime

insolation change. $T_{warm}$ decreases in the northern hemisphere, particularly in the continental mid-latitudes (up to 3.2K). In the southern hemisphere, $T_{warm}$ increases moderately during the Holocene by 0.7K. The signal is reversed in the tropical monsoon areas, probably due to a drop in evaporative cooling in the





northern and an increase in evaporative cooling in the southern hemispheric tropical monsoon regions. Both, $T_{warm}$ and $T_{cold}$ are intercepted by relatively strong 'events' leading to a drop in temperature in all cluster

centres by up to 0.7 for several hundred years. These events may at least partly be associated with the volcanic forcing prescribed to the model. The Pearson correlation coefficient between the detrended $T_{warm}$ cluster centres and the volcanic forcing range from 0.6 (for the increasing cluster centre) to 0.8 (for the decreasing cluster centre). The Pearson correlation coefficient between the detrended $T_{cold}$ cluster centres and the volcanic forcing range from 0.27 (for the decreasing cluster centre) to 0.74 (for the increasing cluster

centre). These events have hardly any effect on the general vegetation trends.

In contrast to the hemispheric uniform temperature trends, the annual mean precipitation ($P_{ann}$) varies at regional level. $P_{ann}$ increases in large parts of North America and in central northern Asia (ca 70-110°E, 45-60°N). In the northern hemispheric monsoon regions, precipitation decreases, particularly in the north African monsoon domain and the southern rim of the Himalayas.

In the southern hemisphere, $P_{ann}$ increases in the monsoon region and the extratropics with the exception of Australia showing a decline in precipitation at the eastern coast and no change in the central and western parts (south of 22°S). The signal in the continental monsoon domains is well in line with the orbital monsoon hypothesis (Kutzbach et al. 1981) that directly links an increase in summer insolation with a strengthening of the summer monsoons and vice versa. $P_{ann}$ also increases in northern Brasil and along the western coasts of

the South American continent and South Africa. Precipitation does not change much north of 60°N, which implies that the shifts in vegetation distribution are rather controlled by temperature changes in this region.

To test the most important climatic driver in the different regions, we performed a constrained ordination via the redundancy analysis function "rda" implemented in the "vegan" package of R (Oksanen et al, 2018).

Fig. 4 summarizes the results, qualitatively showing the main contributors explaining the variance in the

vegetation change and the ratio of explained variance by $T_{warm}$ vs. $P_{ann}$. The temperature of the coldest month explains only a small part of the variance in an initial RDA and was thus excluded in the final RDA.

The RDA result confirms that temperature is the most important variable constraining the vegetation change in the high northern latitudes, but the strong decrease in forest north of 60°N can only be explained by the combined effect of $P_{ann}$ and $T_{warm}$. The shared variance of both variables is higher than the unique contribution

by $T_{warm}$. Also, in most other regions, both variables jointly explain the vegetation dynamics. Variations in $T_{warm}$ are only important in the maritime areas of the northern hemisphere, where changes in vegetation are generally very small (i.e. cluster center shows no change) with the exception of Western Canada.

Precipitation explains most of the slight vegetation change in Eastern Europe and Central Asia and is the most important factor in the southern Sahara region. In the southern hemisphere, the temperature of the

warmest month plays a minor role, variations in $T_{warm}$ explain the vegetation change only in a few, diffusely distributed grid-cells with the exception of Patagonia and Central-West Brasil. Precipitation is the most important driver in large parts of Australia and in the southern hemispheric monsoon margins.

Although the shared variance accounts for the largest part, precipitation in all monsoon regions explains much more of the vegetation change than $T_{warm}$. Given the uniform hemispheric trend in temperatures,

precipitation is the factor that imprints the regional variations on the vegetation signal. We therefore focus on





analysing the precipitation change as the main driver of the global vegetation trend rather than disentangling the temperature signal whose sign is directly forced by the Holocene insolation change. The simulated vegetation distributions for 2.15ka b2k and 8ka b2k are displayed in the Appendix A. Reference maps for the 2.15ka b2k climate states can be found in the Appendix B.

**5. Regional insights into the vegetation changes and the controlling factors of the precipitation change**

**5.1 The northern hemispheric monsoon domain**

Fig. 5 shows the months contributing most to the precipitation change between 8ka and 2.15ka b2k. In both hemispheres, warm season precipitation is the most important driver of the Holocene precipitation signal. For the global monsoon domains, this is well in line with the orbital monsoon hypothesis (Kutzbach, 1981). The rise in northern hemispheric summertime insolation during the early Holocene leads to a strengthening of the thermal gradient between the continents and the oceans, enhancing the monsoon circulation and increasing the summertime precipitation over land. Climate model simulations and reconstructions indicate an increase in moisture availability in the monsoon margin regions and an inland penetration of the vegetation belts (Jolly et al., 1998; Hély et al. 2014; Feng et al., 2006; Zhao et al., 2009, Metcalfe et al., 2015). In the southern hemisphere, the decrease in insolation generally leads to a weakening of the monsoon circulation over land and the opposite effect on precipitation and vegetation (Wang et al. 2014 and references therein).

To assess the change in monsoon strength, we calculated the global monsoon area and the global monsoon precipitation following Zhou et al. (2008) with the modifications of Liu et al. (2009). In this definition, monsoon areas are characterized by a precipitation maximum (>55% of the annual total) in the summer season and a large annual range in rainfall. The summer season includes the months May to September for the northern hemisphere and the months November to March for the southern hemisphere. The reverse is applied for the respective winter season. As 'annual range' the difference between summer and winter precipitation is taken. The monsoon precipitation is given by the sum of total summer rainfall in the monsoon area, taking the latitudinal changes in the grid-cell area into account.

According to our model, the northern hemispheric monsoon region is increased by more than 25% on the continents at 8ka compared to 2.15ka b2k (Tab.3, Fig. 6). The monsoon precipitation is increased by 40% over land. With a northward expansion of about 600 km (up to 22°N), the North African monsoon area is the most enlarged monsoon domain during 8ka b2k. This coincides with a widely spread vegetation into the Sahara at 8ka b2k. In the extended monsoon area, raingreen shrubs are increased by up to 30% (Fig. 7e) and the grass fraction by up to 60%, reflected by the cluster centres with strong decreasing trend over the Holocene (cf. Fig 2). Raingreen shrubs populate the entire western Sahara and reach a coverage of about 10% in large parts, going up to 20% in the coastal regions at 8ka b2k (Fig.7c, Fig A2 in the Appendix A). In the Eastern Sahara they are spread out to only about 24°N. At the northwestern monsoon boundary, the forest cover fraction is decreased during 8ka b2k. This is related to the colder northern hemispheric winter climate during the mid-Holocene leading to unfavourable growth conditions for tropical trees in these grid-cells. The





temperature of the coldest month is around 15.5°C which is a fixed bioclimatic limit for tropical trees in the model. With the continuous boreal winter warming towards PI, cold season temperature more and more exceeds this limit leading to an establishment of tropical deciduous trees. The overall non-vegetated area north of 15°N is diminished by 22% in the mean at 8ka b2k, ranging up to 82% in the Central Sahel.

In the core monsoon domain (0-15°N), total vegetation cover is not increased during the mid-Holocene, but tropical evergreen trees are much more widespread (Fig. 7a). In the course of the Holocene these forested areas are replaced by grassland. Tropical Eastern Africa, experienced an increased total vegetation cover during the mid-Holocene that is determined by an enhanced monsoon related moisture flux convergence in spring (not shown) leading to an expansion of grass covered areas.

In the other northern hemispheric monsoon domains, the vegetation change is more regionally confined. The East Asian monsoon area is expanded to the northwest into Western Mongolia by up to 800km at 8ka compared to 2.15ka b2k (Fig.6). This leads to an increased in total vegetation cover by up to 27% per grid-cell, mainly due to an expansion of the grass cover (Fig.7g). The cover fraction of the extratropical deciduous and evergreen trees is also increased in total, although in several grid boxes it is only one tree PFT being replaced by the other. Pollen based-reconstructions reveal a regionally strongly increased tree cover fraction during the mid-Holocene, particularly in the middle and lower reaches of the Yellow River (Ren, 2007, Tian et al., 2016). This is also the area in which our model indicates the largest changes in tree cover. Over the course of the Holocene, the vegetation belts retreat back to their modern position, coincident with the monsoon retreat, so that the grass fraction at the eastern rim (ca. 117-120°E, 41-46°N) increases towards 2.15ka b2k at the expense of the tree cover fraction (Fig. 7c,d).

The South Asian monsoon is expanded at its northwestern rim by about 1-2 grid-cells (ca. 200km) into western South Asia and onto the Tibetan Plateau (Fig.6), leading to a bare soil fraction decreased by up to 50% at 8ka b2k. The enhanced monsoon coincides with a widespread increase in area covered by raingreen shrubs, extratropical evergreen trees and grassland, in particular on the Tibetan Plateau and along its eastern margin, along the Gulf of Kutch, and in the Thar desert at mid-Holocene (Fig.7). Tropical evergreen forest are decreased along the western coastal region of South Asia due to temperature limitations in the cold season. Our model reveals less vegetation in Central South Asia (17.5-21.5N, 75-82.5E) during the mid-Holocene, driven by a tropical evergreen forest cover decreased by up to 40 % and a C4 grass cover reduced by up to 35%. Partly, these PFTs are replaced by raingreen shrubs and C3 grass at 8ka b2k, reflecting a cooler winter climate. In addition, the model shows rather decreased precipitation in this region at 8ka during mid-summer, caused by a high pressure anomaly in the lower atmosphere above South Asia and Indochina (Fig.8d). The anticyclonic atmospheric flow around this anomaly exhibits low level easterly wind above southern India, inhibiting the moisture flux to Central South Asia (Fig 8c). Furthermore, the high pressure anomaly is associated with strong subsidence (Fig.8a). The monsoon flow stretches more to the north at 8ka compared to 2.15ka b2k. All these factors lead to a decreased moisture availability in this region, worsening the climatic conditions for the establishment of vegetation in the model at mid-Holocene.

The reduction in the South Asian monsoon strength during the Holocene also affect the vegetation on and around the Tibetan Plateau. At 8ka b2k, extratropical evergreen forests are more widespread along the





northeastern and southwestern edge (Fig.7c). On the Tibetan Plateau, grass cover is strongly increased
        compared to 2.15ka b2k (up to 40%), owing to the higher summer precipitation (Fig. 8e).

        The North American monsoon area is expanded south- and westward by approx. 200km at mid-Holocene
        (Fig.6). Summer precipitation is substantially increased above the ocean, in Central America and the north
        western part of the South American continent, leading to a larger area covered by tropical evergreen forests.
In the course of the Holocene, tree cover reduces and is replaced by grass (Fig.7).

### 5.2 The southern hemispheric monsoon domain

        In total, the southern hemispheric monsoon domain is enlarged by 8% and receives 7% more precipitation at
        8ka compared to 2.15ka b2k. However, the continental monsoon area is smaller by 5% and precipitation
level only reach 89% of the 2.15ka values (Tab.3). Thus the changes in the strength of the monsoon during
        the Holocene are much smaller in the southern hemisphere than in the northern hemisphere. For example, the
        monsoon area in South America is reduced by only a few grid cells, mainly in the Western part (Fig.6) at
        mid-Holocene. Nevertheless, annual precipitation in nearly the entire continental monsoon area is lower at
        8ka than at 2.15ka b2k (Fig.3), due to a generally northward displaced Intertropical Convergence Zone and
diminished uplift of moist air over the central continent during the monsoon season (cf. Fig. 9). Both is also
        mirrored in the slightly increased precipitation on the north-easternmost continent (i.e. Northeastern Brazil
        and the Caribbean region). This pattern is supported by reconstructions and other simulations (Maksic et al.
        2018; Prado et al. 2013a and b, Cruz et al. 2009) for the mid-Holocene.

        The reduced precipitation has a minor effect on the total vegetation cover that is only slightly decreased
along the foot of the Bolivian Andes. The vegetation in the Amazon rainforest region is relatively stable, but
        forest is shrunk at its southern edge during the mid-Holocene, confirming the picture revealed by
        reconstructions (e.g. Mayle et al., 2000, Mayle and Power, 2008, Rossetti et al, 2017). Outside of this region,
        it is rather the vegetation composition that changes. The cover fraction of tropical evergreen trees is
        increased by up to 86% in a broad band in southeastern Brazil (Fig.7a), which is overcompensated by a
diminished fraction of extratropical evergreen trees and partly also extratropical deciduous trees. Thus, the
        total tree cover in this region is decreased and replaced by grassland at 8ka compared to 2.15ka b2k,
        indicating a decreased Atlantic rainforest and more open vegetation similar as proposed in reconstructions
        (e.g. Ledru et al., 2009). In the model, this is the effect of a climate characterised by warmer winters, but that
        is still not moisture-limited in the monsoon domain.

On the central continent (ca. 15-30°S, along 60°W), extratropical and tropical evergreen tree covers are
        diminished by up to 22% leading to less forested area during the mid-Holocene. In this region, grass cover is
        also slightly enhanced during 8ka b2k. This pattern of forest transitions fits well to a recent palaeo-record
        synthesis revealing the most significant Holocene transitions in vegetation for southwestern Amazonia and
        southeastern Brazil, induced by a less intense South American summer monsoon (cf. Smith and Mayle,
2018).



The precipitation on the South African continent is diminished in the entire monsoon domain during the mid-Holocene austral summer (Fig. 9e) as well as the annual mean (Fig.3). This can mostly be attributed to the less powerful monsoon circulation at 8ka compared to 2.15ka b2k. The Angola low is weakened and the South Atlantic high is shifted northwards (Fig. 9d), reducing the moisture flux to the continent and thereby decreasing precipitation (cf. Vigaud et al. 2009). The relatively weaker South Indian high induces an anticyclonic circulation with a northerly wind anomaly along the coast and landward low-level winds (Fig. 9c). This leads to increased precipitation over the ocean and decreased precipitation over land at 8ka b2k. The ITCZ is weakened, coinciding with reduced uplift (Fig. 9a) and inhibited convection.

The South African monsoon area is reduced by about 200km in its northwestern part (ca. 3.5-5.5°S, 14-24°E) at 8ka b2k (Fig.6). This is related to the fact that the West African monsoon dynamics also imprint on the Tropics in the southern hemisphere (up to ca. 10°S), probably due to easterly moisture-bearing trade winds off the Indian Ocean (Burrough and Thomas, 2013; Vincens et al., 2005). As a result, the tropical region not only experiences an increase in summer precipitation due to the intensification of the South African monsoon over the course of the Holocene, but also a decrease in winter precipitation due to the weakening of the West African monsoon and easterly flux from the Indian Ocean (Fig.8). Therefore the annual range of precipitation in this region is reduced during the mid-Holocene, not fulfilling the criteria of monsoon areas used in this study.

The overall result of the change in precipitation is an increased cover fraction of tropical evergreen trees (up to 16%) in parts of Central (12-20°S, 20-29°E) and Eastern (12-14S, 31-36°E) Africa at 8ka b2k (Fig.7a). Over the course of the Holocene, tree cover reduces and is replaced by grassland, in line with the tropical regions on the north African continent. South of 15°S, the vegetation change is more complex. The total vegetation is decreased in three north-south orientated stripes, mainly due to a decreased grass cover at 8ka b2k. East of 30°E, extratropical trees are reduced during the mid-Holocene and partly replaced by grassland. West of 30°E tropical evergreen trees strongly recede over the course of the Holocene and give way to raingreen shrubs and C4 grass. This is probably also related to the bioclimatic limit for the temperature of the coldest month needed for the establishment of tropical trees in the model (Tab.1), that is at some point in time no longer met in the cooling winter climate in the southern hemisphere during the Holocene. Southern Africa experiences rather an increase in C3 grass and extratropical trees from 8ka to 2.15ka b2k, while raingreen shrubs slightly decrease (Fig.7e).

Reconstructions show a diverse picture, but rather show more humid vegetation types at mid-Holocene and aridification afterwards (Olago, 2001; Burrough and Thomas, 2013 and references therein). This may at least partly be related to the fact that the vegetation distribution in Southern Africa is strongly controlled by fires, maintaining grassland and savannas despite sufficient rainfall for the establishment of forest (Bond et al., 2003, Bond, 2008). JSBACH includes a fire model, but the effect of fires may be underestimated in the Southern African region. At least, Southern African vegetation seems to be highly correlated with climate in the our simulation.

The Australian monsoon region is shrunk at its south-eastern rim by approx. 200km at 8ka b2k (Fig.6). The high pressure anomaly above the continent leads to an anticyclonic flow with low-level westerly winds south



of the monsoon domain (ca. 22°S) and easterly winds in the monsoon region (Fig.9c). This results in a

weakened monsoon-related moisture flux and less precipitation during austral summer at mid-Holocene (Fig.9e). The total vegetation is decreased by up to 19% during the mid-Holocene, mainly due to a decreased C4 grass cover and less tropical evergreen trees. On the southern edge of the monsoon domain, the tropical evergreen tree cover is raised by up to 20% at the expense of extratropical evergreen trees and raingreen shrubs (Fig.7). This switch between different woody PFTs is mainly determined by the warmer southern

hemispheric winter climate during mid-Holocene.

### 5.3 Extratropical North America

According to the model, the total vegetation cover in large parts of North America increases from 8ka to 2.15ka b2k (Fig.2). Most prominent is the vast expansion of trees, ranging from the southern Great Plains to

the Canadian interior plains and the northeastern Rockies that is also seen in reconstructions (Grimm et al., 2001). In the model, this is manifested by a rise in the extratropical evergreen tree fraction by up to 24% west of 100°W and a rise in extratropical deciduous tree fraction by up to 45% east of 100°W (Fig.7). At mid-Holocene, most of this region is covered by grassland or bare soil. In the southern Great Plains, raingreen shrubs are more widespread at 8ka, these are replaced by grass during the Holocene.

The precipitation response to the orbital forcing in the model is a complex mixture of several interacting processes. In total, the Great Plains receive less precipitation during the mid-Holocene (Fig.3a). Fig. 5 shows that the summer months contribute most to the annual mean precipitation change. These are also the months in which the northern hemispheric monsoon dynamics are most vigorous and the orbital forcing has the strongest imprint on the monsoon circulation. Summer monsoon precipitation coincides with a strong

condensational latent heat release, affecting also the tropospheric circulation in the extratropics. The subtropical anticyclones are stated to be related to Kelvin and Rossby wave responses to the heating in the monsoon rainband (Matsuno 1966, Gill 1980, Rodwell and Hoskins, 2001, Trenberth et al., 2000). These anticyclones cover about 40% of the Earth surface (Rodwell and Hoskins, 2001) and are therefore a major player in the global circulation of the atmosphere and oceans and the atmospheric teleconnection pattern.

This suggests that the strong changes in the northern hemispheric monsoon systems also play a decisive role in modifying the extratropical vegetation change over the course of the Holocene. Fig. 8d displays the mean change in the atmospheric standing waves during June, July and August (JJA). The Bermuda high is strengthened in the core at 8ka b2k leading to a rerouting of the moisture transport from the Gulf of Mexico along the Atlantic coast, thereby enhancing the precipitation along the Appalachian mountain range (cf. also

Williams et al., 2010). In this region of raised precipitation levels, the extratropical deciduous tree fraction and with it the total forest cover is even higher during the mid-Holocene than at 2.15ka b2k.

The moisture flux is more divergent in the Great Plains and the atmosphere is generally drier, inhibiting convection and rainfall. The intensity change of the Bermuda high has so far been seen as one possible explanation for the change in Holocene vegetation pattern seen in pollen-based reconstructions (e.g. Grimm

et al., 2001 and references therein). In addition, the model shows a rainfall-reducing subsidence anomaly



over the Great Plains south of 45°N at 8ka b2k (Fig.8a) that could at least partly be attributed to a Rodwell&Hoskins-like Rossby wave response to the enhanced updrafts in the North American summer monsoon (Harrison et al., 2003). The northward shifted upper level westerly Jet during the mid-Holocene (Fig.8b) coincides with a northward replacement of the storm tracks, which probably results in less transient

eddy transport to the northern Great Plains at 8ka b2k (Seager et al., 2014). The model also reveals less precipitation during spring and therefore limited evaporation during mid-Holocene summer (not shown), reducing water recycling and the northward transport of moisture in the Great Plain low level Jet. The influence of evaporation on vegetation change may also explain the relatively large proportion of vegetation variance explained by the temperature of the warmest month (cf. Fig.4).

The more La Nina like sea surface temperature pattern (Fig. B1, Appendix B) may also contribute to the drier surface conditions in the Great Plains during mid-Holocene, although the pattern is probably less pronounced in the model than indicated by reconstructions (Shin et al, 2006). Which of these mechanism is actually the main driver in the Holocene precipitation change can not be disentangled in the model without performing a complex set of sensitivity experiments.

The reduced precipitation above the Canadian interior Plains and northeastern Rockies (ca. 120-90°W, 45-60°N) during the mid-Holocene, accompanying the decreased extratropical evergreen tree fraction there (Fig.7c), is probably related to the strengthened and westward shifted North Pacific subtropical high (Fig.8d). This leads to enhanced northerly winds transporting rather dry continental air inland. The subsidence is increased in large parts during mid-Holocene summers (Fig.8a), further limiting the

convection. At least partly, the differences in subsidence can be contributed to the change in dynamics, coinciding with the northward displaced westerly jet at 8ka b2k. This also affects the region along the northern Rockies and Alaska (45-60°N, east of 120°W). Here, the westward shift in the north Pacific High reduce the moisture flux to the area, resulting in less moisture convergence in the region. Therefore precipitation is decreased at 8ka b2k in the annual mean that leads to less total vegetation, mainly due to a

reduced grass cover (up to 35%) at mid-Holocene. The fraction of extratropical evergreen trees decreases over the course of the Holocene and is replaced by cold-resistant shrubs and grassland (Fig.7), probably reflecting the typical Holocene signal in the Taiga-Tundra margin.

## 5.4 Taiga-Tundra dynamics in the high northern latitudes

The taiga-tundra region and the dependence of the vegetation signal on temperature changes has been widely discussed before (e.g. Bigelow et al., 2013, MacDonald et al. 2000, Wanner et al., 2008). Our model is in line with previous results and shows no new insights regarding the long-term trend. Therefore, this region will not be explored in detail in this study. The main signal is the decrease of tundra along the Arctic coast and the strong decrease of boreal forest further south, both reflecting the southward shift of the boreal vegetation

zones over the course of the Holocene (Fig.2 and Fig.7), also found in previous transient vegetation simulations (e.g. Braconnot et al., 2019). The main forcing is the cooling of the summer climate and the warming of the winter climate during the Holocene (Fig.3) that is induced by the seasonal changes in





insolation. In North America, the boreal forest area is reduced by 25.5% and the mean northern treeline (here
ad hoc defined as isoline of a zonal mean tree cover fraction of 10%) moves approx. 2° of latitudes to the
south over the course of the Holocene (Fig.7). Europe experiences less change. The boreal forest area shrinks
by 5.9% and the mean northern treeline shifts southward by approx. 0.5° of latitudes. In Asia, the boreal tree
cover fraction reduces by nearly one third from 8ka to 2.15ka b2k and the northern treeline shifts southward
by approx. 3° of latitudes. The boreal trees are mainly replaced by C3 grass and cold-resistant shrubs.

However, two regions stand out due to an increase in the bare soil fraction during the Holocene. These are an
area in mid-Siberia (about 100-140°E, north of 58°N) and a region in northern Canada (about 100-120°W,
north of 60°N). The increase may be a model specific response. During the mid-Holocene, modelled winter
temperatures are too cold to allow the establishment of extratropical evergreen trees in these regions (limited
by a mean temperature of the coldest month below -32.5°C). With the increase in wintertime insolation and a
successive winter warming over the Holocene, this limit no longer applies, and extratropical evergreen trees
replace the extratropical deciduous trees (Fig.7). However, the net primary productivity (NPP) of the
evergreen trees is too low to build up their living tissues completely. This, by definition, increases the
simulated bare soil fraction.

In another inland region in Siberia (ca. 70-110°, 48-58°N), the increase in forest cover south of the modern
treeline is disproportionately higher than in the other regions. This is in line with a substantial increase in
precipitation in this region (Fig.3) over the course of the Holocene. At mid-Holocene, the westerly wind jet is
shifted to the north and is weaker (Fig.8b). The coincident subsidence results in decreased summer
precipitation at 8ka b2k, similar to the changes in Canada.

In line with the results by Braconnot et al. (2019), the worldwide strongest changes in tree cover fraction
occur north of 60°N, while tree cover changes in the rest of Eurasia are rather small.


## 5.5 Extramonsoonal Australia and South America

The northern part of Australia (north of 22°S) is characterized by an increase in total vegetation during the
Holocene related to the strengthening of the Australian monsoon (cf. 4.2.2). South of 22°S vegetation
decreases during the Holocene (Fig.2a). At 8ka b2k, extratropical evergreen trees are more widespread.
Along the eastern coast, where proxy records and our model reveal wetter mid-Holocene conditions (e.g. Mc
Glone et al., 1992), the cover fraction of extratropical forests is increased by up to 16% (Fig.7). In central
Australia, forest fraction is raised by up to 6%. The grass cover is enhanced in large parts of the (today)
rather dry continental interior. Over the course of the Holocene, these grasslands and forested area are partly
replaced by raingreen shrubs.

These vegetation changes are related to a slightly wetter climate during the mid-Holocene. In large parts of
the continent, the precipitation difference between 8ka and 2.15ka b2k are largest during November
(continental interior, Fig.5), October (eastern coast) and June (parts of the southern coast). During these
months, the subtropical ridge is weaker at 8ka b2k (also seen in Fig. 10d). This leads to less subsidence (Fig.
10a) and favours convection and precipitation (Fig.10c). In addition, the moisture influx from the western



Pacific is increased. The enhanced low pressure system over northwestern Australia during austral spring causes monsoon-like conditions and suggest an earlier onset of the Australian summer monsoon at 8ka b2k, coinciding with an enhanced precipitation level in entire Australia. The warmer sea surface temperatures in the Indonesian ocean may additionally favour increased winter precipitation during mid-Holocene (Verdon and Franks, 2005). Furthermore, a more La Nina sea surface pattern in the tropical Pacific is known to

increase moisture level in arid Australia (Nicholls, 1992; Quigley et al., 2010, Barr et al., 2019).

The Gran Chaco and Pampas regions east of the South American Andes experience an increase in vegetation during the Holocene, mainly due to an increase in grass cover (Fig.7). On the Southern continent, extratropical evergreen trees cover up to 27% less area at 8ka compared to 2.15ka b2k. These vegetation changes are probably related to the increase in both, austral wintertime and summertime precipitation during

the Holocene. Large parts of the region get precipitation mainly due to the moisture influx in the north-easterly branch of the Subtropical Atlantic High. During 8ka b2k austral winters, this anticyclone is shifted equatorwards leading to diverging easterly wind anomalies in the lower level and a decrease in moisture flux convergence (not shown). The upper tropospheric westerly jet is squeezed and intensified along the core (ca.30°S), a pattern that is underlined by Holocene reconstructions (Lamy et al., 2010). The changes in

Westerlies probably enhance the subsidence in the lee of the Andes. The subsidence is furthermore enhanced by the updraft of the Afro-Asian monsoon that leads to a strong divergence anomaly in the upper tropospheric velocity potential (Fig. 11) and accordingly to a strong convergence anomaly above South America during austral winter, triggering subsidence. Fig.5 indicates, that the precipitation difference between 8ka and 2.15ka b2k in this region is largest during the month November and December. During

austral summer, a deep continental low evolves above the southern American continent that additionally deflects the easterly winds to the south so that the wind is channeled between the Brazilian Plateau and the Andes. This low level jet transports huge amounts of moisture southwards at present (Marengo et al., 2004; Garreaud et al., 2009). During mid-Holocene Nov. and Dec., this low is weakened, in line with the weakened South American monsoon (Fig.9), whose outflow touches this region at 2.15ka but not at 8ka b2k. The model

indicate a southwind anomaly in the lower troposphere (i.e. a reduced low level jet), diminishing the inflow from the monsoon area (not shown).

**6. Are the Holocene vegetation changes as linear as they appear in our global analysis?**

The overview of the patterns of clustered vegetation trends reveals rather linear transitions between different PFT groups during the Holocene (Fig.2). This appears to be consistent with the assumption that large-scale

vegetation change and climate change directly follow the subtle, approximately linear orbital forcing over the last millennia. However, reconstructions also report of rapid, or abrupt, vegetation changes in different regions of the world. The most prominent example is the expansion of the Sahara from a rather 'green Sahara' to its current extent some 5500 years ago. Indicators of regionally (mainly in the western part) rapid changes include the rapid increase in dust deposition found in marine sediments core off the Western Sahara

coast (e.g. deMenocal et al, 2000). Abrupt regional vegetation changes, at least for individual taxa, are also suggested to exist in Central Asia (Zhao et al., 2017), the high northern latitudes (MacDonald et al. 2000),





parts of Northern America (e.g. Marsicek et al., 2013; Shuman et al., 2009; Foster et al, 2006) and Europe (Giesecke et al., 2011, Seddon et al., 2015). How rapid these changes were and whether these changes only reflect local phenomena is still a matter of discussion. Mayewski et al. (2004) identified several phases with
rapid global climate changes during the Holocene which might have forced changes in vegetation composition. Ecosystems can respond quite slowly to changes in the local environmental conditions, but can also change rapidly when the conditions approach critical thresholds (Scheffer et al., 2001). In addition, initially slow changes can be amplified by feedbacks in the climate system, giving rise to non-linear responses to the orbital forcing (Williams et al., 2002).

The cluster method used to group the globally-wide trend in main PFTs in this study is not designed to properly identify rapid changes. The aggregation of the PFTs in main PFT groups complicates the occurrence of rapid changes as these changes would have to be very extensive and accompanied by very strong climatic changes for an entire PFT group to collapse. Furthermore, the global view masks regionally appearing transitions. To detect non-linearity in the Holocene vegetation change, we suggest evaluating the relative
change ($R$) in the individual PFT fractions (see Appendix C). The change in the individual PFT fraction between two consecutive time-steps is compared to the maximum possible change during the period of interest. The noise time-series are filtered by a Butterworth-Filter with a cut-off frequency $1/F$ = 500 years. As measure for rapid changes, the maximum $R_{max}$ of the absolute value of $R$ is taken.

Fig.12 shows the global patterns of $R_{max}$ for the individual PFTs. In total, six regions can be identified that
indicate rapid changes in PFT cover fractions, i.e. shifts that exceed the linear trend by a factor of up to 5, given the low pass filter of 500 years. These regions include areas in the high northern latitudes and the monsoon margins in each hemisphere. The areas coincide with regions where the cluster method also shows relatively strong changes, except for the region of South America in which the rapid changes occur within the forest PFTs.

In northern Canada and eastern Siberia (region 1 and 2), rapid shifts mainly occur between the extratropical deciduous trees and extratropical evergreen trees, and, to a weaker extent, in the bare soil fraction. The main processes for the transition are the same as discussed above in section 5.4. A typical example of such a transition in region 1 and 2 is depicted in Fig 13a for a grid box in eastern Siberia. During the first millennia, extratropical evergreen trees cannot exist, because the temperature of the coldest month is too low. This
bioclimatic limit mimics the dependence of extratropical trees on moisture in the high latitudes, because there are no moisture-related bioclimatic thresholds defined in JSBACH. In a very cold, and thus dry, climate, the deciduous trees have an advantage over evergreen trees. Extratropical evergreen trees and extratropical deciduous trees can coexist after about 6 ka b2k. The time scale of the transition from around 6 ka b2k to some 3.5 ka b2k is governed by the small difference in NPP between the tree PFTs. During the last
millennium, rapid fluctuations within a few decades between extratropical evergreen and deciduous trees are found in the simulations. These are induced by fluctuations in the temperature of the coldest month, which may fall below the bioclimatic limit at which extratropical evergreen trees can exist in some years. The extratropical evergreen trees decay within a few decades, and this decay is governed by the time scale of mortality of 50 years prescribed in JSBACH. The extratropical evergreen trees eventually recover at a slower



pace which follows the difference in NPP development and which is not explicitly determined by the time scales prescribed in JSBACH.

In the Sahel-Sahara transition zone (region 3), the model reveals rapid changes which are also found in some, but not all, pollen records (e.g. Kröpelin et al., 2008, Shanahan et al., 2015) or in Saharan dust transport (e.g. deMenocal et al., 2000). The latter change can directly be linked to rapid landscape changes mainly in the

western part of the Sahara (Egerer et al., 2016). The simulated bare soil fraction rapidly increases at the expense of grass cover and raingreen shrubs. In some grid cells simulated tropical evergreen trees are rapidly substituted by tropical deciduous trees. Both changes can be interpreted as a non-linear response to the steady decline in moisture level caused by the retreating summer monsoon during the Holocene. While the shift between tropical tree PFTs can be rapid, the amplitude of these changes is rather small. A more detailed

discussion of simulated and reconstructed vegetation changes in the Sahara-Sahel region is in preparation for a subsequent study.

In of South Asia and in parts of Southeast Asia (region 4), various combinations of rapid changes are found in the simulation. In South Asia, raingreen shrubs decrease rapidly and are replaced by tropical evergreen trees and grasses. As an example, the vegetation dynamics of one grid cell from this region are depicted in

Fig. 13b. The transition between the different PFTs is discussed in section 5.1. As in the high northern latitudes, the pace of the transition at the mid-Holocene is governed by the differences in NPP of the PFTs involved. The time scales of allocation and mortality of the competing PFTs differ. They range from 30 years for tropical trees to 12 years for raingreen shrubs. The bioclimatic limit which restricts the occurrence of tropical trees is imposed by the temperature of the coldest month (see section 5.1). In the time period from

around 7 ka b2k to 5.5 ka b2k this limit is frequently crossed. Because the time scales of allocation and mortality are shorter than that of the PFTs at high northern latitudes, the change between PFTs appears to be more rapid, and the amplitude is larger.

The climatic changes that drive the rapid swings in PFTs in the South American region (region 5) and in the South African region (region 6) are discussed in section 5.2. The governing time scales of PFT dynamics and

the bioclimatic limits involved are the same as for tropical Asia (region 4).

## 7 Summary and Conclusion

Using a cluster analysis technique on a Holocene simulation for the period 8ka to 2.15ka b2k with MPI-ESM, the main global natural vegetation trends during the Holocene are identified and discussed in the

context of the climate change over the Holocene. A redundancy analysis reveals that precipitation is the main climatic driver leading to regional differences in the vegetation change outside the high northern latitudes. The model is well in line with the overall trends reported by pollen-based vegetation reconstructions. These main trends in the model are:

a) The southward retreat of the tundra and northern treeline by up to 3° of latitudes, expressed by a bipolar

change pattern for all PFT groups with a decrease on the northern site and an increase further south. Asia faces the major change in the model by a reduction of the boreal tree cover by nearly one third since the mid-





Holocene. This transition is mainly related to the seasonal temperature trend driven by the change in insolation during the Holocene. Boreal winter temperatures rise, summer temperatures decline, reducing the growing season and stressing the vegetation in the high northern latitudes. In the polar regions, climate

becomes even too harsh for vegetation to survive.

b) A decrease and equator-ward retreat of all vegetation types in the northern hemispheric monsoon regions, leading to an increase in the deserts, particularly to a vast expansion of the Sahara. The mean non-vegetated area in North Africa north of 15°N increases by 22% during the Holocene, ranging up to 82% in southern Niger. The reduction in vegetation coincides with the equator-ward retreat and the weakening of the

northern-hemispheric monsoon belts. Our model indicates a decrease of the continental northern hemispheric monsoon area by approx. 25% since 8ka b2k. The precipitation in the monsoon regions decreases by approx. 40% over land, leading to limited moisture availability in large regions.

c) An increase in the forest and partly the grass fraction in extratropical North America, leading to an overall increase in vegetated area during the Holocene, forced by a rising moisture level. From the model simulation

we find several possible mechanisms triggering the Holocene increase in precipitation in North America. The North Pacific high is shifted westward and the westerly jet and the related storm tracks are shifted northward at mid-Holocene, leading to less moisture transport to Canada and the northern Great Plains and enhanced subsidence in large parts of the region. The Bermuda high is slightly strengthened during 8ka summer, rerouting the moisture transport along the coast and reducing the moisture flux convergence over the Great

Plains. Additionally, the strengthening of the ascent in the North American summer monsoon is probably responsible for enhanced subsidence in the Great Plains.

d) A small increase in vegetation in the southern hemispheric monsoon regions, induced by an increase in forest, grass and shrubs (S Africa). Due to the decreased austral summer insolation during the mid-Holocene, the southern hemispheric monsoon systems are weaker at 8ka compared to 2.15ka b2k. The continental

monsoon domain is reduced by 5% and the precipitation is decreased by 11%. Hence, the change in the monsoon strength is weaker than for the northern hemisphere. This is also reflected in the relatively minor changes in total vegetation cover. Instead, the southern hemispheric monsoon regions experiences (strong) shifts in the woody PFT composition. The warmer winter climate during the mid-Holocene allows for the establishment of tropical evergreen trees in the model. In the course of the gradual cooling during the

Holocene, winter temperatures at some point fall outside the tolerance range of tropical trees, which give way for other woody PFTs. Under the relatively humid conditions in South America, extratropical trees win this competition. In South Africa, where the moisture is more limited, evergreen trees are replaced by raingreen shrubs. Australia experiences both, an increase in extratropical trees (mainly at the east coast) and in raingreen shrubs.

e) An increase in the bare soil fraction in extra-monsoonal Australia and an increase in grass and total vegetation in extra-monsoonal South America. Mid-Holocene Australian climate (south of 22°S) is slightly wetter than at 2.15ka b2k due to a weaker subtropical ridge, less subsidence and a stronger moisture influx from the Pacific Ocean. During the Holocene, precipitation declines, leading to a decrease of the grass cover and extratropical trees that can not be overcompensated by the establishing raingreen shrubs. In South



America, austral wintertime precipitation is decreased at the mid-Holocene due to increased subsidence and a decrease in moisture flux convergence caused by the poleward shift of the South Atlantic subtropical high. At least partly, the stronger subsidence is related to a remote response to the enhanced North African summer monsoon. In addition, austral summer precipitation is decreased, probably due to a weaker influence of the South American monsoon, upstream of the main lower tropospheric wind field. The increase in rainfall

during the Holocene leads to an increase in grass cover (Gran Chaco and Pampas region) and extratropical evergreen trees (Eastern Argentinia).

At large scales, the global shifts in vegetation appear to follow the slow, to first order linear, orbital forcing. In some regions, however, the model simulates rapid, strongly nonlinear changes in PFTs. These regions include two domains in the high northern latitudes (Canada and mid-Siberia), and four regions in the

monsoon margin area on both hemispheres (Sahel-Sahara, India & South East Asia, South America and South Africa). The rapid change in bare soil fraction in the Sahara region can be attributed to the nonlinear response of the monsoon precipitation and vegetation to the orbital forcing (cf. Dallmeyer et al., 2020).

Transitions between individual PFTs triggered by changes in climate appear at a large spectrum of time scales. The transitions are governed by the difference in NPP of the competing PFTs. If the PFTs have the

same time scales of allocation and mortality, then transitions can take place over many centuries. More rapid transitions in PFTs can occur within a few decades or less, if the climate moves in or out the bioclimatic tolerance so that some temperature thresholds in the model are suddenly met or not fulfilled any more. These rapid changes in simulated vegetation pattern have to be interpreted with care. Bioclimatic limits are known to exist for plant species. Whether the concept of bioclimatic limits can be applied to the temporal dynamics

of PFTs has to be critically re-assessed. PFTs encompass a variety of different and diverse plant species. Because of this plant diversity within a PFT, PFTs presumably show a larger resilience to climate changes than individual species do. Hence our model results should be taken as an indicator of possible rapid changes in the vegetation, rather than a precise prediction of rapid changes themselves.

This study explores the Holocene vegetation changes around the world and interprets them in terms of the Holocene climate change. The model results reveal that most of the Holocene vegetation trends seen outside the high northern latitudes can be attributed to modifications in the intensity of the global summer monsoons. Due to the seasonal changes in insolation, the monsoon systems in the northern hemisphere weaken, while the monsoon systems in the southern hemisphere intensify during the Holocene. Due to teleconnections

(Rodwell-Hoskins-response to the diabatic heating), the intensity change of the monsoons affect the subtropical anticyclones, whose modification in turn leads to an adjustment of the wind field over the extratropical continents. In addition, the magnitude of the updrafts in the monsoon belts determines the strength of the subsidence in the extra-monsoonal areas, such that a weakening of the monsoon rainfall coinciding ascent also weakens the descent abroad. The model results thus identify the global monsoon

system as the key player in Holocene climate and vegetation history and point to a far greater importance of the monsoon systems on the extra-monsoonal regions than previously assumed. The model results indicate



that changes in the global monsoon dynamic should always be considered as an actor in past and future climate changes, even in regions not primarily influenced by monsoons.





**Appendix A: Simulated vegetation distributions for the 2.15ka b2k and the 8ka b2ka time-slices**

**Appendix B: simulated climate at 2.15ka**

**Appendix C: Definition of the relative change $R$**

We evaluate the temporal change $dV/dt$ of the simulated vegetation fraction $V$ of a PFT relative to an overall

amplitude $\Delta V$ and over the entire period $\Delta T$ considered, in our case, $\Delta T = 6000$ y. We define the relative change $R$ by

$$R = \left(\frac{dV}{dt}\right)\left(\frac{\Delta V}{\Delta T}\right)^{-1}.$$

We define the overall amplitude $\Delta V$ as the maximum possible change that the vegetation fraction can reach, i.e. $\Delta V = 1$, to focus on large and rapid vegetation changes. For the same reason, we use a Butterworth filter

of the order 5 to further separate fluctuations in simulated vegetation fraction which occur on all time scales. Figure C1 demonstrates the effect of filtering. In Figure C1, changes in extratropical deciduous trees in a grid box in East Siberia are depicted. The unfiltered time series reveals small, rapid changes at decadal time scale in the first millennia of the simulation, a large transition around 4000 ka b2k, and larger and rapid decadal fluctuations during the last millennia. When applying a Butterworth filter of order 5 with a cut-off frequency

$1/F > 1/200$y, then only the large transition around 4000 ka b2k is identified as the most rapid change. Figure 12 shows the maxima of the absolute value of $R$ using $1/F = 1/500$y,

$$R_{max} = \left.\left|\frac{dV}{dt}\right|\right|_{max} \left(\frac{|\Delta V|}{\Delta T}\right)^{-1},$$

hence we do not differentiate cases of strong increase or decrease in vegetation fraction. Using smaller or larger values of $F$ would result in different values of $R_{max}$, as indicated in the example Fig.C1, but the overall

global patterns, specifically the regions of strongest non-linear change shown in Fig.12, would be more or less the same.



**Code availability**

**Data availability**

The primary data, i.e. the model code for MPI-ESM, are freely available to the scientific community and can be accessed with a license. In addition, secondary data and scripts that may be useful in reproducing the authors' work are archived by the Max Planck Institute for Meteorology and are accessible without any restrictions (link will be provided at the final submission).

The Biome6000 pollen-based biome reconstructions (Harrison, 2017) that we used for the evaluation of the model can be downloaded from: https://doi.org/10.17864/1947.99.

The volcanic forcing data will be made available at https://doi.pangaea.de/10.1594/PANGAEA.928646.

**Author contribution**

AD and MC wrote the manuscript, SL ran the simulation, MS and MT provided the volcanic forcing data. 855 MC, UH and AD planed the study and were involved in the analysis. All authors discussed the analysis and the manuscript.

**Competing interests**

The authors declare that they have no conflict of interests.

**Acknowledgements**

This work contributes to the project PalMod, funded by the German Federal Ministry of Education and Research (BMBF), Research for Sustainability initiative (FONA, www.fona.de). AD was financed by PalMod (Grant number: 01LP1920A). MS acknowledges funding from the European Research Council (ERC) under the European Union's Horizon 2020 research and innovation programme (grant agreement No 820047). We thank T. Kleinen (MPI-M) for his helpful comments on an earlier version of this manuscript. We acknowledge J. McConnell, J. Cole-Dai and M. Severi 865 for providing ice-core data.



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





| No. | landcover classification | phenology type | $Tc_{min}$ [°C] | $Tc_{max}$ [°C] | $Tw_{max}$ [°C] | GDD5 [°C] | time scale[y] |
|---|---|---|---|---|---|---|---|
| 1 | tropical evergreen trees | raingreen | 15.5 | - | - | 0 | 30 |
| 2 | tropical deciduous trees | raingreen | 15.5 | - | - | 0 | 30 |
| 3 | extratrop. evergreen trees | evergreen | -32.5 | 18.5 | - | 400 | 60 |
| 4 | extratrop. deciduous trees | summergreen | - | 18.5 | - | 400 | 60 |
| 5 | raingreen shrubs | raingreen | 2.0 | - | - | 900 | 12 |
| 6 | cold shrubs | summergreen | - | -2 | 18 | 350 | 24 |
| 7 | C3 grass | grasses | - | 15 | - | 0 | 1 |
| 8 | C4 grass | grasses | 10 | - | - | 0 | 1 |

1215

Table.1: Bioclimatic limits and time-scale of allocation and mortality for the eight natural plant functional types (PFTs) used in the dynamic vegetation module of the model JSBACH. Listed are phenology type, PFT-specific minimum ($Tc_{min}$) and maximum ($Tc_{max}$) of the mean temperature of the coldest month, PFT-specific maximum mean temperature ($Tw_{max}$) of the warmest month, and growing degree days, i.e. temperature sum of days with temperatures exceeding 5°C (GDD5). All temperature values are given in °C.

| Biome | BNS |
|---|---|
| tropical forest | 0.58 |
| warm-temperate mixed forest | 0.51 |
| temperate forest | 0.88 |
| boreal forest | 0.79 |
| savanna and dry woodland | 0.12 |
| grassland and dry shrubland | 0.63 |
| desert | 0.39 |
| tundra | 0.53 |
| total | 0.73 |

Table 2: Best neighbour score (BNS) showing the agreement between pollen-based reconstructions and the mega-biomes converted from the simulated plant functional type cover fractions. Values between 0.2 and 0.5 means 'fair to good' agreement, values above 0.5 'good to very good' agreement, and above 0.8 reveal an 'excellent' agreement. For details on the metric see Dallmeyer et al. (2019).

70





| Region | | Monsoon precipitation | Monsoon area |
|--------|--------|-----------------------|--------------|
| NH | land | +40 % | + 25 % |
| | total | + 6 % | + 4 % |
| SH | land | - 11 % | - 5 % |
| | total | + 7 % | + 8 % |

Table 3: Simulated change in northern hemispheric (NH) and southern hemispheric (SH) monsoon area and monsoon precipitation for the mid-Holocene (8ka b2k) compared to the late-Holocene (2.15ka b2k). The changes have been calculated, following the method of Zhou et al. (2008) with the modifications of Liu et al. (2009).

Figure 1: Simulated (a) and reconstructed (b) biomes for the 6ka time-slice and the simulated (c) and reconstructed (d) change in vegetation between 6ka and pre-industrial (PI), expressed in terms of the ecological development. To assess the biome change, the biomes has been further grouped into the main categories forest, savannas, grassland/tundra and desert. Positive ecology development describes the transition from biome indicating more open landscape to biomes indicating less open landscape (e.g. desert in 6ka to grasslands or to savannas or to forests in PI). Negative development describe the opposite transition (e.g. from grassland in 6ka to desert in PI).

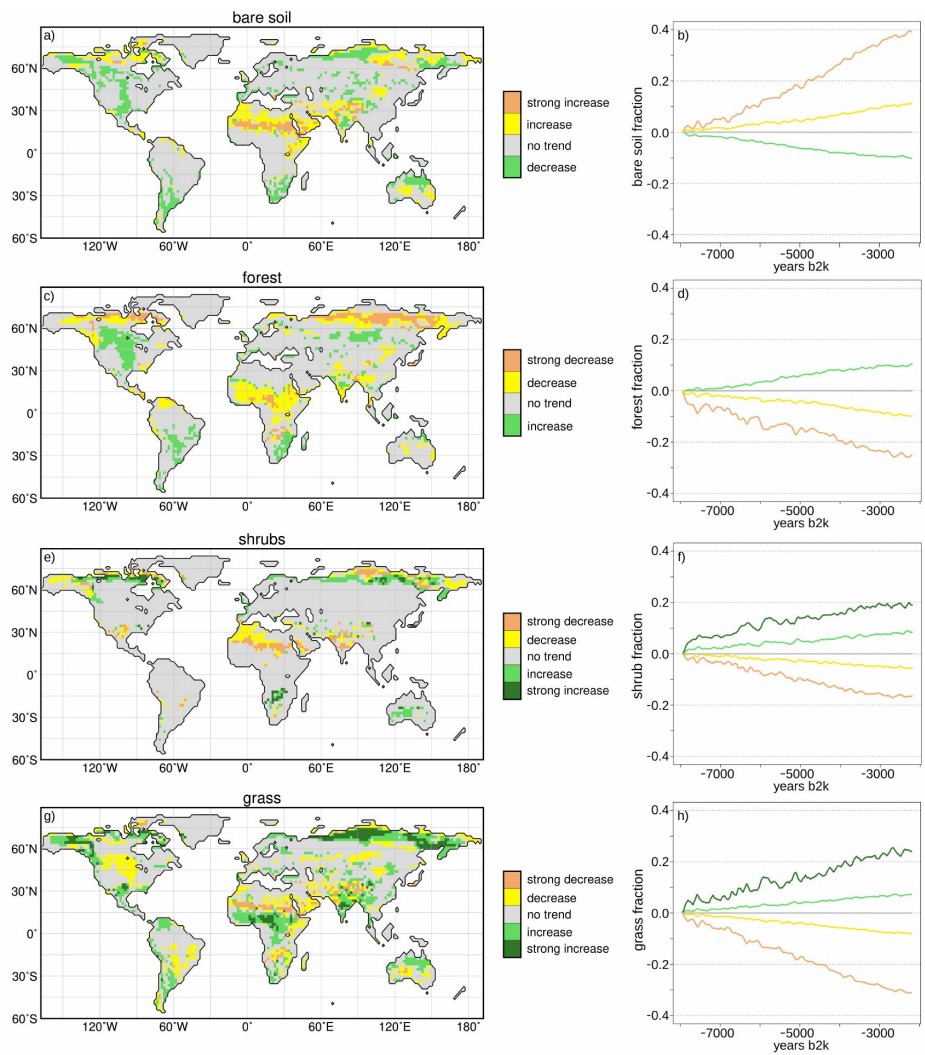

Figure 2: Simulated Holocene vegetation cluster, derived by c-means clustering method (R, Meyer et al. 2017). The left panel displays the cluster pattern for (a) the bare soil cover, which is 1 minus the total vegetation in the model, (c) forest cover, (e) shrub cover, and (g) grass cover. The right panel shows the respective trends in the cluster centres, i.e. for (b) bare soil fraction, (d) forest fraction, (f) shrub fraction, and (h) grass fraction. For the analysis, 100 year running means of the anomaly time-series (fraction at certain time-step – 8ka b2k climatological mean) have been taken. The cover fraction change of the cluster centres can range from -1 to 1 in the model, i.e. decrease of 100% or increase of 100%, respectively.

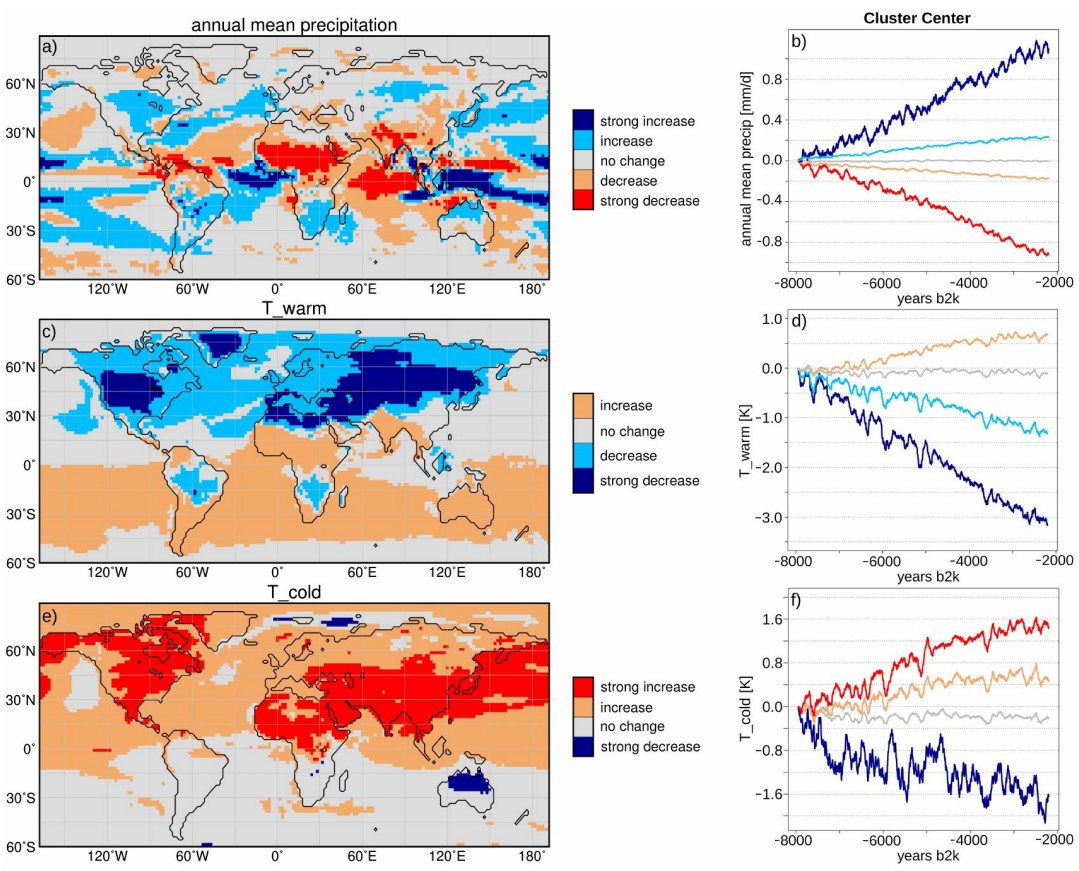

Figure 3: Simulated Holocene climate cluster, derived by c-means clustering method (R, Meyer et al., 2017). The left panel displays the cluster pattern for (a) annual mean precipitation, (c) temperature of the warmest month (T_warm), and (e) temperature of the coldest month (T_cold). The right panel shows the respective trends in the cluster centres, i.e. for (b) annual mean precipitation [mm/ d], (d) T_warm [K], and (f) T_cold [K]. For the analysis, 100 year running means of the anomaly time-series (value at certain time-step – 8ka b2k climatological mean) have been taken.



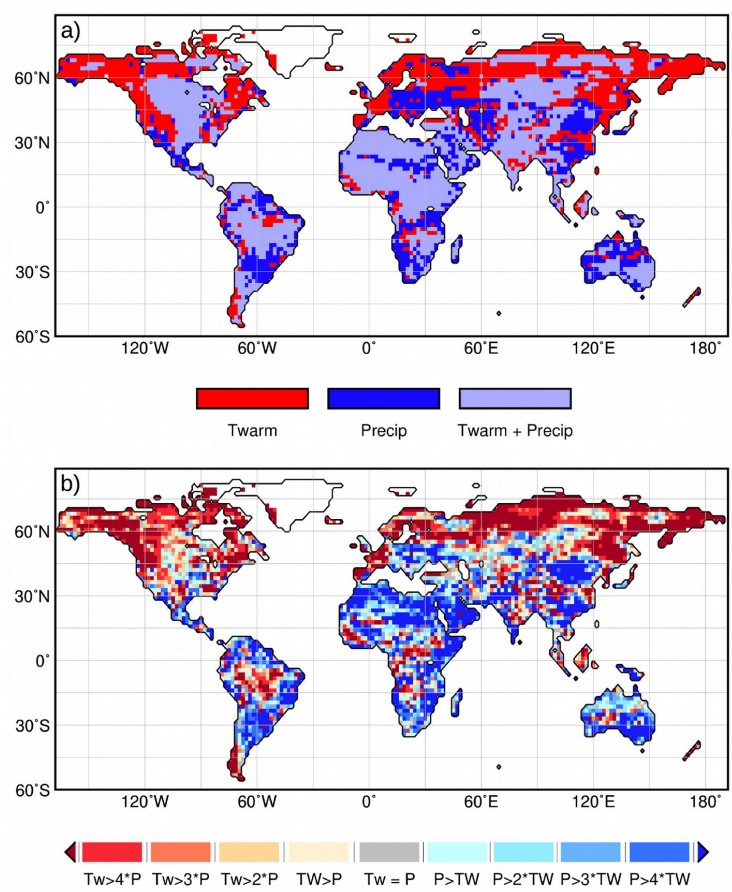

Figure 4: Results of the redundancy analyses performed by the VEGAN routine in R (Oksanen et al, 2018) for the vegetation groups forest, shrubs, and grass, with precipitation (precip) and temperature of the warmest month (Twarm) as explanatory variables. a) displays the regions in which the variance in the Holocene vegetation change is mostly explained by the variance in Twarm (red), or in precipitation (blue) or in the shared variance of Twarm and P; b) shows the ratio of the variance explained by Twarm (Tw) and by precipitation (P). For instance, P>4*Tw means that precipitation explains more than 4 times more of the variance than Twarm.



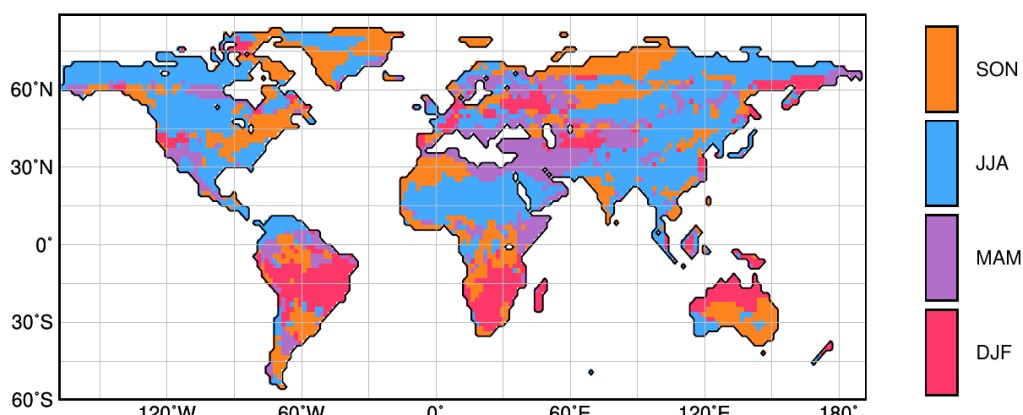

Figure 5: Season which contributes most to the annual mean precipitation change between 8ka b2k and 2.15ka b2k in the model, i.e. September to November, June - August (JJA), March to May (MAM) and December to February (DJF). We are aware of the fact, that the length of the season changes during the Holocene due to the precessional cycle (c.f. Bartlein and Shafer, 2018). However, this does not change the response of the atmospheric physics in the model to the insolation change, so we assume that the results are not affected by a fixed calendar. But please keep in mind, that we define the seasons according to the modern calendar implemented in the model, i.e. any statements regarding seasonal changes only refer to the simulation and might not necessarily be transferable to the real Holocene seasonal changes.

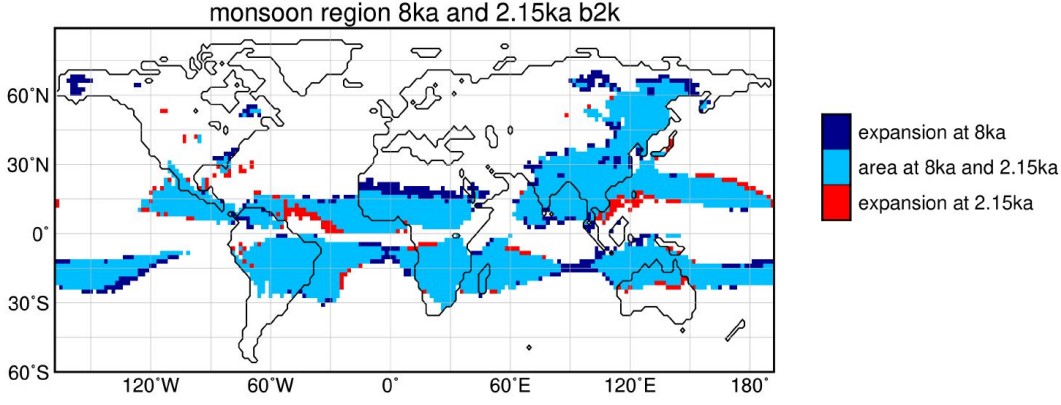

Figure 6: Monsoon region at 8ka and 2.15ka b2k, derived by the method by Zhou et al. (2008) with the modifications by Liu et al. (2009). The figure shows the monsoon area that is indicated for both time period, i.e. does not change from mid-to late Holocene (light blue), area additionally assigned to the monsoon domain at 8ka b2k, but not at 2.15 ka b2k (dark blue) and area additionally assigned to the monsoon domain at 2.15 ka b2k, but not at 8ka b2k.



Figure 7: Simulated changes in plant function types (PFT) for the mid-Holocene time-slice (8ka b2k) compared to the late-Holocene (2.15ka b2k), i.e. yellowish colour indicate a decreased in PFT cover fraction at 8ka b2k, greenish colour an increased PFT cover fraction at 8ka b2k compared to 2.15ka b2k. Values of '1.0' mean that the grid-box is fully covered by the PFT during 8ka b2k, but does not occur at 2.15ka b2k in this grid-cell, and vice versa for values of '-1.0'.



Figure 8: Simulated difference in June to August (JJA) mean climate in the model for the time-slice 8ka b2k compared to 2.15ka b2k, i.e. a) vertical velocity (omega) in 500hPa [$10^{-2}$ m/s] with blue colour showing increased uplift during 8ka b2k; b) upper-tropospheric zonal wind (u) in 250hPa [m/s] with blue colours indicating enhanced easterly wind; c) low-level zonal flow (u) in 850hPa [m/s]; d) zonal anomaly in lower tropospheric streamfunction (psi*, in $10^7$ m²/s) reflecting the change in standing waves, with blue colours indicating increased cyclonic circulation at 8ka b2k compared to 2.15ka b2k; and e) mean seasonal precipitation [mm/d] with blue colours showing enhanced precipitation during 8ka b2k.

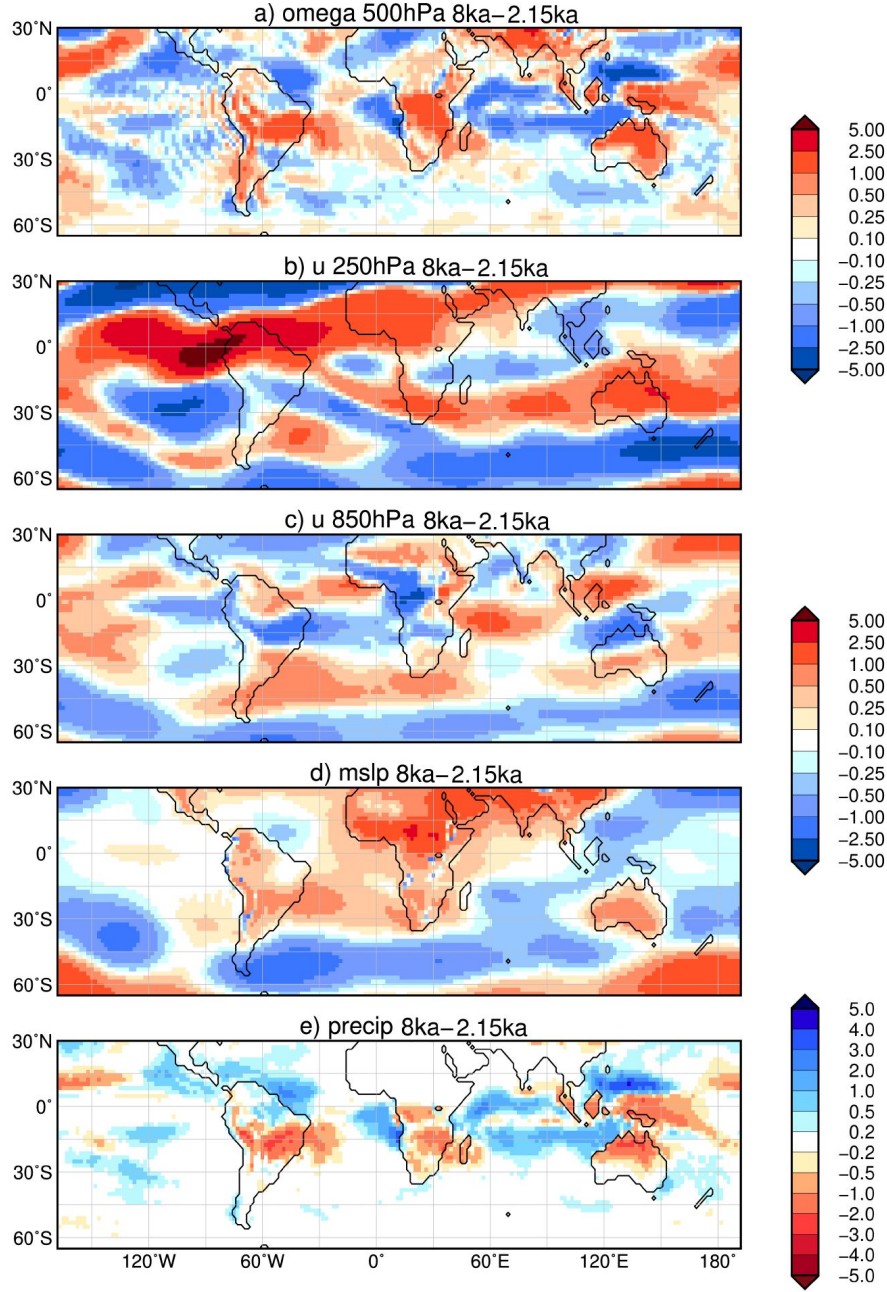

Figure 9: Simulated difference in December to February (DJF, austral summer) mean climate in the model for the time-slice 8ka b2k compared to 2.15ka b2k, i.e. a) vertical velocity (omega) in 500hPa [$10^{-2}$ m/s] with blue colour showing increased uplift during 8ka b2k; b) upper-tropospheric zonal wind (u) in 250hPa [m/s] with blue colours indicating enhanced easterly wind; c) low-level zonal flow (u) in 850hPa [m/s]; d) mean sea level pressure [hPa] with low pressure anomaly given in blue colours; and e) mean seasonal precipitation [mm/d] with blue colours showing enhanced precipitation during 8ka b2k.





Figure 10: Simulated difference in September to November (SON) mean climate in the model for the time-slice 8ka b2k compared to 2.15ka b2k, i.e. a) vertical velocity (omega) in 500hPa [$10^{-2}$ m/s] with blue colour showing increased uplift during 8ka b2k; b) upper-tropospheric zonal wind (u) in 250hPa [m/s] with blue colours indicating enhanced easterly wind; c) low-level zonal flow (u) in 850hPa [m/s]; d) zonal anomaly in lower tropospheric streamfunction (psi*, in $10^{7}$ m²/s) reflecting the change in standing waves, with blue colours indicating increased cyclonic circulation at 8ka b2k compared to 2.15ka b2k; and e) mean seasonal precipitation [mm/d] with blue colours showing enhanced precipitation during 8ka b2k.



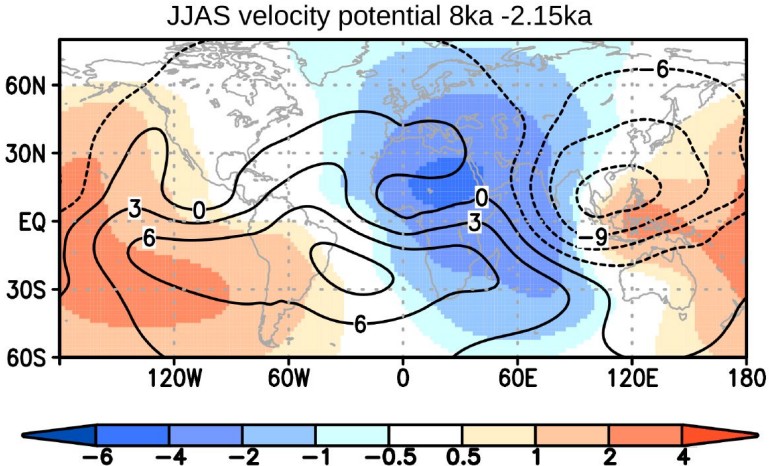

Figure 11: Simulated difference (shaded) in June to September mean upper tropospheric (250hPa) velocity potential [km$^2$/s] between the time-slice 8ka b2k and 2.15ka b2k. Bluish colour indicate enhanced divergence, coinciding vertical ascent in the troposphere below, redish color indicate stronger convergence, coincinding subsidence below. The contours plotted on top show the respective velocity potential at 2.15ka b2k. Negative values display divergence, positive values convergence.



Figure 12: maximum relative change in PFTs (R), indicating the rapidness of the PFT change. The more yellowish the more rapid is the transition. The model reveals six different regions with rapid changes (region 1-6).





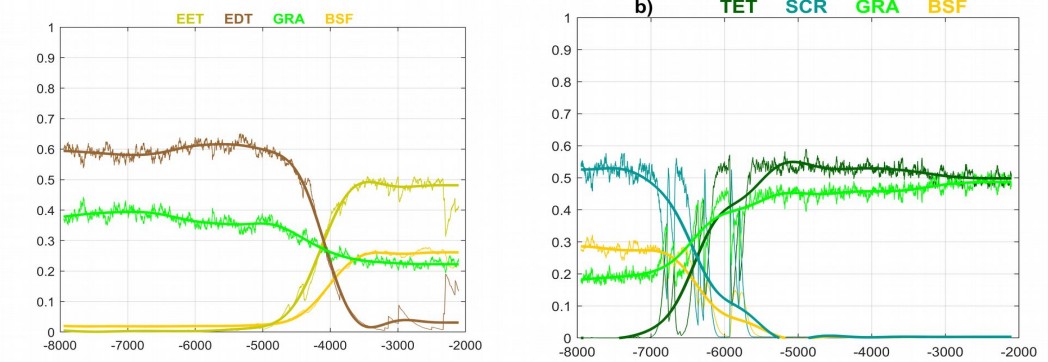

Figure 13: Examples of rapid transitions between simulated PFTs in a grid cell in East Siberia (left, region 2 in Fig. 12) and in South Asia (right, region 4 in fig. 12). The PFTs are EET: extratropical evergreeen trees, EDT: extratropcial deciduous trees, TET: tropical evergreen trees, SCR: shrubs (in this location, rain-green shrubs), GRA: grass, BSF: bare soil fraction. The thin lines are annual values of fractional vegetation coverage, and the thick lines are filtered time series using a Butterworth filter of order 5 with a low-pass filter of 500 years.





Figure A1: Simulated plant functional type distributions at 2.15ka b2k.





Figure A2: Simulated plant functional type distributions at 8ka b2k.

100

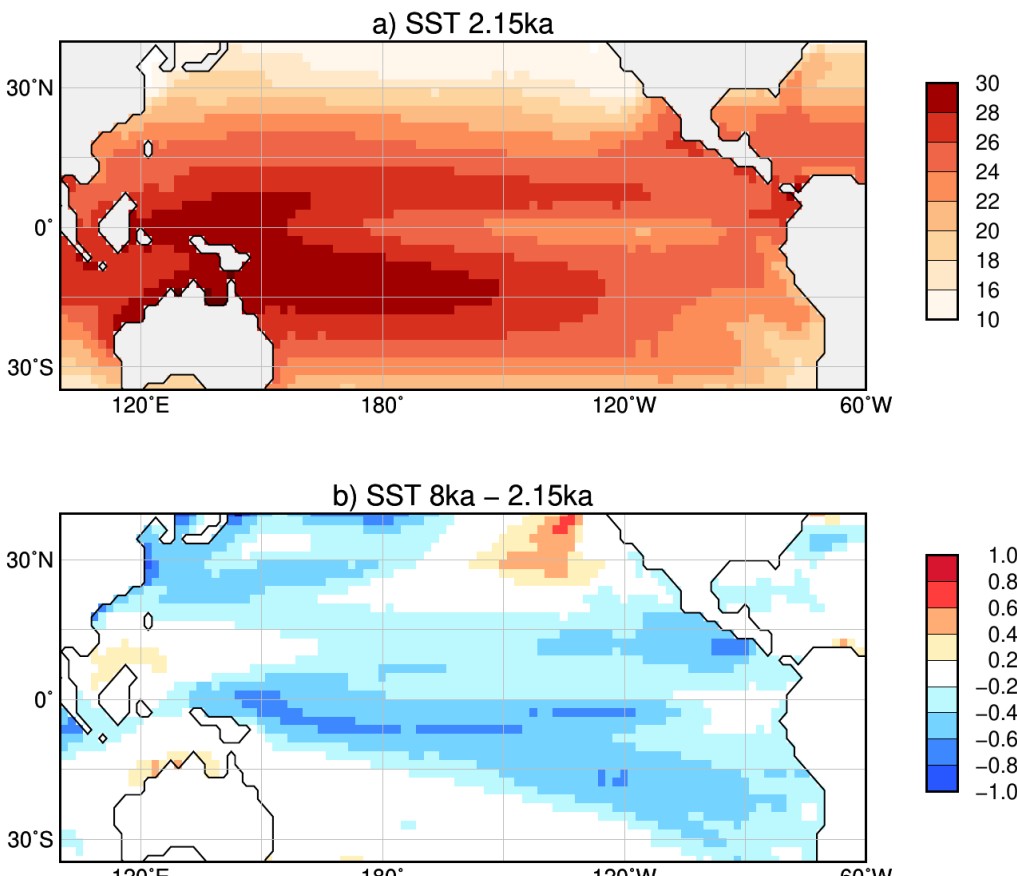

Figure B1: Simulated annual mean sea surface temperatures (SST) for a) the 2.15ka b2k time-slice and b) the difference between 8ka and 2.15ka b2k.

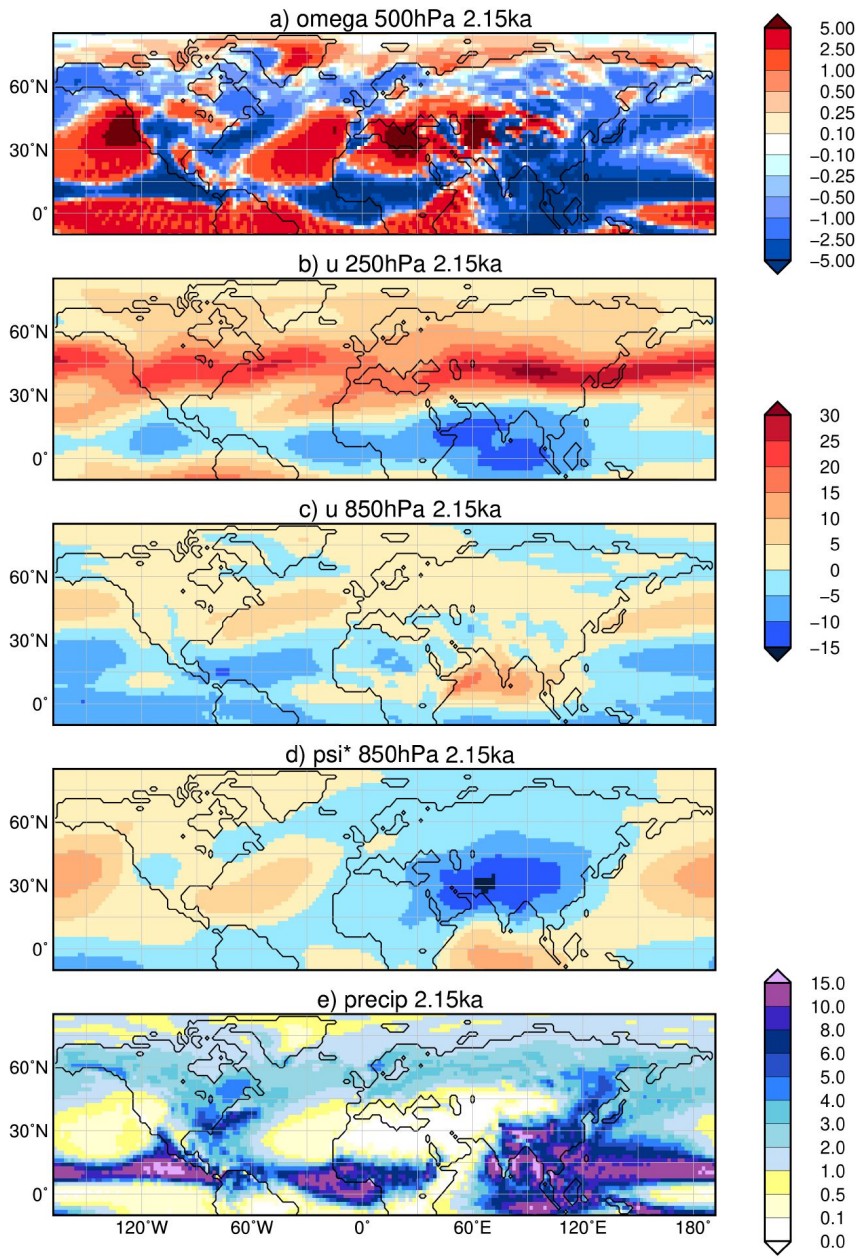

Figure B2: Simulated JJA mean climate at 2.15ka b2k, i.e. a) vertical velocity (omega) in 500hPa [$10^{-2}$ m/s] with blue colour showing uplift; b) upper-tropospheric zonal wind (u) in 250hPa [m/s] with blue colours indicating easterly wind; c) low-level zonal flow (u) in 850hPa [m/s]; d) zonal anomaly in lower tropospheric streamfunction (psi*, in $10^{7}$ m²/s) reflecting the standing waves, with blue colours indicating cyclonic circulation; and e) mean seasonal precipitation [mm/d]

Figure B3: Simulated SON mean climate at 2.15ka b2k, i.e. a) vertical velocity (omega) in 500hPa [10⁻² m/s] with blue colour showing uplift; b) upper-tropospheric zonal wind (u) in 250hPa [m/s] with blue colours indicating easterly wind; c) low-level zonal flow (u) in 850hPa [m/s]; d) mean sea level pressure anomaly [hPa] to standard atmospheric mean pressure (here 1013.25hPa), with blue colours indicating low-pressure systems; and e) mean seasonal precipitation [mm/d].

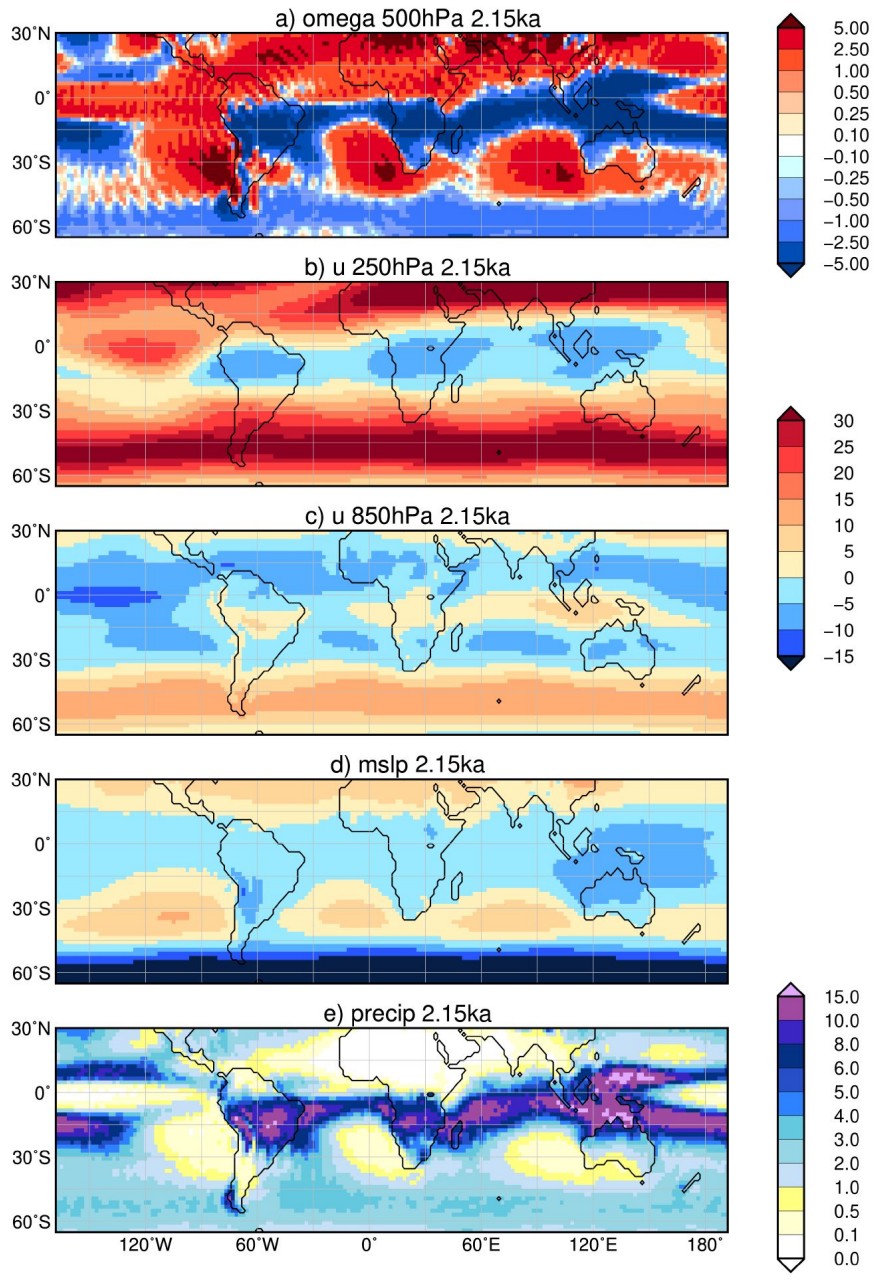

Figure B4: Simulated DJF mean climate at 2.15ka b2k, i.e. a) vertical velocity (omega) in 500hPa [$10^{-2}$ m/s] with blue colour showing uplift; b) upper-tropospheric zonal wind (u) in 250hPa [m/s] with blue colours indicating easterly wind; c) low-level zonal flow (u) in 850hPa [m/s]; d) mean sea level pressure anomaly [hPa] to standard atmospheric mean pressure (here 1013.25hPa), with blue colours indicating low-pressure systems; and e) mean seasonal precipitation [mm/d].



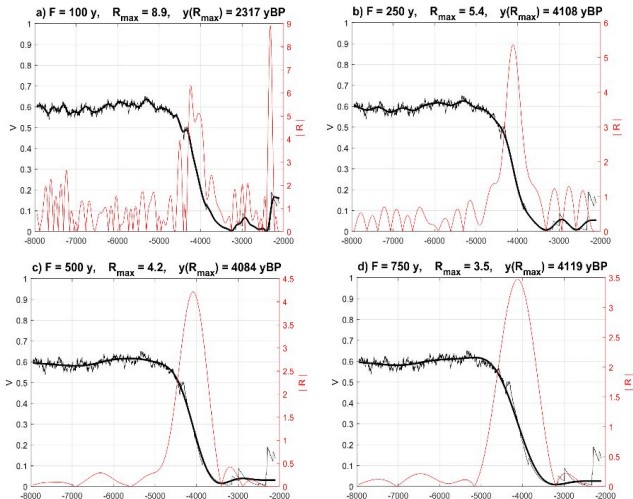

Figure C1: effect of filtering on Rmax.