# Peer review of "Holocene vegetation transitions and their climatic drivers in MPI-ESM1.2"

_Climate of the Past, 2021_

## Referee Comment (RC2)

Figure C1: effect of filtering on Rmax.

110

[referee-annotated manuscript omitted]

---

## Author Comment (AC1)

**Reply to Referee comments on CP-2021-51, "Holocene vegetation transitions and their climatic drivers in MPI-ESM1.2" by Anne Dallmeyer et al.**

RC1:

We thank the referee Qiong Zhang for her constructive and valuable comments on our manuscript and her suggestions for improving the results.

In the following, we respond to her comments (typed in black) point by point. Our responses are marked in blue.

R: Dear Anne and co-authors,

I have read through your submitted manuscript, noticed it is the work you presented in EGU 2021, a really nice work. In this manuscript you have presented the results from a 8000-year long Holocene transient simulation with MPI-ESM1.2, focus on the long term trend in different vegetation type and changes in global and regional vegetation pattern. The simulated vegetation changes are compared with the pollen-based reconstructions and showed good agreement. The climatic drivers for these changes are identified using redundancy analysis, you conclude that the overall trend in global pattern is linearly following the orbital forcing, and some rapid non-linear changes are observed in a few regions - not only the well-know Sahara region but also in northern high latitudes and other monsoon margins. You also found that the precipitation is the main driver for regional difference in northern high latitudes, and the mechanism is associated to the changes in circulation that induced by the changes in global summer monsoon.

The manuscript is logically structured and very well written, has provided comprehensive information regarding the changes in vegetation both in MPI-ESM1.2 model simulation and in pollen reconstructions. Such detailed information on the trend and changes in global and regional vegetation pattern certainly contribute a good reference for both the climate modelling community and paleo-climate community. I have a few comments below for you to consider to improve the presentation.

A: Thank you for pointing to the scientific importance of our manuscript.

R: I understand that this long transient simulation is performed from 6000 BCE to 2000 CE for 8000 years. In section 2.3 (p7-line234) the authors explain that for an easier nomenclature they define mid-Holocene time slice to 8 ka b2k. However, it is not easy for most of us who always regard mid-Holocene as 6 ka, as showed in the literature in the introduction. When reading the manuscript I often have to remind myself this 8 ka b2k actually is the commonly mentioned 6ka, and have to do some convert when looking at the figures. I suggest to use conventional 6 ka in the paper, some readers may misunderstand (if not read carefully but only take a look at figures) that 8 ka here refers to 2000 years before 6ka. This manuscript focuses on the natural vegetation variability, and the presented results do not include the last 2000 years (past2K),

therefore present 6000 BCE (6ka) to 0 would nicely fall to the common understanding.

A: We see this point. Most time-slice studies have been prepared for the 6ka period and our study deals with the vegetation change from 8ka until 2.15ka. Since the vegetation change is rather linear, it can be expected, that the results and conclusions derived in this study do not change considering the 6ka time-slice instead of the 8ka time-slice. The absolute values in the vegetation change will of course be reduced, but not the sign and the order of magnitude of the signal. We will therefore keep the 8ka time-slice, but to reduce the confusion, we will name it "early mid-Holocene", instead of just "mid-Holocene".

R: For the different between the mid and late Holocene, you compute the difference for first 100 years and last 100 years without land-use (250-150 BCE). Considering that the simulation shows the multi-centennial variability (as mentioned in P4-line132), I suggest to use few more hundreds years' data to compute the change between the mid and late Holocene, or check the dominant frequency, I notice that for butterworth-Filter you used 500 years. This can remove the possible impact of multi-centennial variability. Besides, when presenting the changes between two periods, should provide the statistical significance test to show the changes are robust.

A: We decided to use 'only' 100 years because it is the commonly used period for climatological means and to be consistent to the cluster-analysis that is also based on 100-year running means. Since the simulation is transient, longer periods may be affected by trends.
We have tested the significance of the change in vegetation cover and climate by a paired t-test. The difference in vegetation shown before in Fig.7 are all significant, but please notice, that JSBACH uses partly multi-year climatological means to calculate the vegetation. Therefore the variability is expected to be small. We will revise the climate plots (Fig.8,9,10) that they will only show significant changes. Considering only significant changes does not change any results and conclusions presented in the original manuscript.

R: The changes in regional vegetation pattern are presented extensively in section 5, it would be nice to have a schematic diagram or table to summarise the major conclusion (vegetation type with changed percentage etc.) instead of long summary text that somewhat repeat section 5.

A: Thank you for bringing this up. We will include a summarising sketch of the Holocene vegetation change for the readers' convenience, but we decided to keep additionally the summary text. We agree that this may be longer than usual, but it was important to us to take up the five main regions (a-e) listed in chapter 4 in the summary again and to connect them with the summarized results from chapter 5.

R: It is interesting to see that there are more regions showing rapid vegetation shift besides the well-know collapse of green Sahara, the examples given in Figure 13 showing the rapid transition are impressive. However, the two grid box showed in Fig.13 seems they are selective examples, I would expect to see the transition features in Sahara first, also expect to see transitions in more PFTs as showed in Fig.12.

A: We appreciate Qiong's suggestion, and we are going to extend the text on the transition in the Sahara/Sahel region considerably. In addition, we will add a new sub-figure to Fig. 13 which depicts the transition between grass coverage to bare soil and the transition between tropical evergreen to deciduous trees. We will also add a new Fig. 14 which shows the transition in the Sahara/Sahel over a larger region.

R: P6-line 195, a "spin-down" is used, can you explain why use this instead of "spin-up"?

A:We agree that the term "spin-down" may be confusing for most readers. The reason for using this term was that the mid-Holocene equilibrium simulation was branched off the pre-industrial control run and ran somehow 'back in time' until mid-Holocene conditions have been approached. To distinguish this method technically from the spin-up of the PI control run, we called this "spin-down". To avoid this confusion we change "spin-down" to "spin-up" in the revised manuscript.

R: P10-line 335, mentioned "These events may at least partly be associated with the volcanic forcing prescribed to the model", more clarification would be good, does it mean those events do not present in the non-volcanic forced simulation in this model? A reference on this could help the understanding.

A: The volcanic forcing affects the climate response and contributes to the climate variability. In the vegetation change, this forcing is rather buffered, related to the model structure and equations. The analysis of the volcanic impact would go beyond this paper, but will be the subject of another study, prepared by our colleagues.

R: P11-line 396, "monsoon region is increased" better as "monsoon area is expanded".
A: okay.

R: P18, line 661, " last millennia", should be "last 8000 years"
A: okay

---

## Author Comment (AC2)

**Reply to Referee comments on CP-2021-51, "Holocene vegetation transitions and their climatic drivers in MPI-ESM1.2" by Anne Dallmeyer et al.**

**RC2**

We would like to thank Referee #2 for his/her constructive and comprehensive suggestion to improve the manuscript. The Referee's comments are marked in black, our response in blue.

R: This is a solid paper that presents a new set of transient climate simulations for the last 6,000 years, using a state-of-the-art Earth System Model. The paper presents an extensive series of model results and data-model comparisons. That said, it is not clear what new has been learned by this paper, nor how it advances on prior work.

A: Our study is the first, detailed overview of global vegetation transitions during the Holocene. We agree to the Referee that, at least with respect to reconstructions, many individual studies exist discussing the local, regional, or even ecotone-wide vegetation changes, but the vegetation changes in the global context have rarely been discussed. These reconstruction-based analyses assume climatic drivers. Since we use a comprehensive Earth System Model, we are able to analyse a consistent vegetation-climate system and therefore we are able to discuss the atmospheric circulation that goes in line with the vegetation change. To further stress the advances on prior work, we will revise the Introduction and describe the objectives of the study more sharply (see our response below on the more specified comments on the Introduction)

R: The paper's presentation in its current form is highly descriptive, with lengthy results sections (4-6) and really no discussion section to contextualize these new findings in light of the prior literature.

A: We agree with the Referee that section 5 is rather descriptive. This is because we describe the vegetation changes in the five cluster identified in section 4 in more detail and discuss them in the context of the atmospheric circulation. This analysis includes a comparison to previous studies. One aim of our study is to give a complete overview on the Holocene vegetation changes. Therefore we consider it important to include this section 5 with the detailed discussion of the vegetation changes and the climate drivers and do not see a possibility to considerably shorten this chapter, unless we reduce the discussed regions, which in our opinion would degrade the quality of the paper. Following the suggestion by Referee #1, we will add a summarising overview sketch on the vegetation changes and the climate drivers for the readers' convenience.

R: Most of the work on biome comparisons and monsoon dynamics was well done, but it was hard to see what was new here relative to prior papers.

A: Neither the biome comparison nor the analysis of the monsoon dynamics are stated to be the main topic of our study. The study is on the vegetation change. The analysis of the biome change (evaluation of the results) and of the monsoon dynamic intend to understand the vegetation change. With respect to the monsoon dynamic, we did not find any new results, but we have not claimed this at any point. We will sharpen our introduction for clarification and to avoid false expectation.

R: I thought the analysis of linear and abrupt responses was perhaps the most interesting and novel part of this paper. Understood that bioclimatic thresholds as represented by this model may not be the true driver of abrupt vegetation change, but still this paper presents interesting hypotheses about where and when we would expect abrupt vegetation change given climatic drivers.

A: According to the suggestions by Qiong (Referee #1), we will extend this interesting part on non-linear vegetation changes and extend the discussion on the vegetation transitions in the Sahara/Sahel region considerably.

My comments are organized into Major Comments, Line-by-Line comments, and minor annotations on the PDF. The PDF annotations are just grammatical, not substantive.

Other major comments:

R: The Introduction covers the right papers (mostly, see below), but comes across as an unfocused region-by-region review of the paleovegetation and paleoclimatic literature. The Introduction doesn't really set up the key topics that the Roadmap Paragraph (L130-142) establishes as paper foci: abrupt vegetation change; biome-scale vegetation changes; climate drivers. We get a sense that a lot of prior work has been done, but not where there are key knowledge gaps or unresolved questions that this paper will help address.

A: The aim of our Introduction is to give an overview on the important climate and vegetation changes in the different regions of the world and to introduce the main "actors". We agree that this lengthens the introduction, but as the reconstructed vegetation changes during the Holocene are diverse, regional vegetation changes have to be mentioned. We consider it a strength of the paper that we give a complete overview of global vegetation change and do not exclude regions. This is also emphasized by Referee #1 who stated: "*Such detailed information on the trend and changes in global and regional vegetation pattern certainly contribute a good reference for both the climate modelling community and paleo-climate community.*" In order to stress this more clearly we will revise the introduction and sharpen the research questions and will stress the unique features of the study more clearly. We will furthermore try to shorten the introduction.

R: Also, the Roadmap Paragraph does a good job of telling me what analyses will be done, but doesn't really explain what questions will be asked and answered, nor what hypotheses will be tested. What are the overall goal(s) of this paper: Test MPI-ESM1.2 against proxy data? Gain a better understanding of climatic drivers of Holocene vegetation? Feedbacks? Causes of abrupt vegetation change?

A: We agree to the Referee that the aims of the study and the research questions have not been worked out clearly in the original manuscript. We will change this in the revised version.

R: Similarly, clarify in Intro/Roadmap what is new about this study. The latest MPI-ESM1.2 paper is 2019, but the latest vegetation paper is 2013, so is the new contribution the simulation of a fully transient and coupled vegetation-climate model for the Holocene? Given that at least three other papers have been published by this team with this transient simulation (see L199-200), how does this paper advance beyond these prior works?

A: This paper is based on a new transient simulation with improved forcing. The simulation has not been used before. The studies cited in L199-200 deal with completely other topics, they are not about vegetation, although biogeographic dynamics are generally included in the MPI-ESM simulations.

R: The abrupt vegetation change analyses (Section 6, Figs. 12, 13) are to me the most interesting part of the paper. They essentially represent a model hypothesis that most abrupt climate change is forced by PFTs passing bioclimatic limits at the edge of their distribution.  It would be interesting to compare this to the global rate of change vegetation analyses just published by (Mottl et al., 2021).  (Note that this paper had a correction just published, so I recommend contacting the authors to confirm which version of the data to use.)  These analyses also leverage the strength of this paper, which is its transient simulations for the last 6000 yrs.

A: We appreciate very much that the reviewer shared with us the publication by Mottl et al. (2021), which appeared after we submitted our manuscript to Climate of the Past. Our results agree with the results shown in Mottl et al.: In the time period between 8ky BP and around 4 ky BP, the large-scale patterns tend to change linearly and steadily. After some 4 ky BP, the data in Mottl et al. (2021) tend to change more rapidly, presumably because of early land use. At this point, we cannot compare our results with that of Mottl et al., because we have focused our analysis on natural vegetation transition, but not on the effects of land use. Hence, we intentionally disregarded the last 2000 years for which land use is prescribed to expand in our model. During these 2000 years, we, in fact,  see a number of grid boxes - mainly in the old world - in which natural vegetation coverage changes rapidly because of the prescribed conversion of natural land to agriculture or pasture in our model.

A: Appendix C needs more description; the figure looks interesting but the legend is short and cryptic.

R: We agree with reviewer 2 that the caption to Figure C1 is - due to oversight - cryptic. A comprehensive caption will be added. Likewise, we are going to extend the description in Appendix C.

R: Section 5 is rather long and descriptive. Suggest shortening substantially.

A: We agree with the Referee that section 5 is rather long and descriptive. It contains a detailed analysis of the vegetation and climate changes in the different "centres of changes" inferred in the cluster analysis. As Referee #1 emphasize, one of the strength of the paper is the complete overview on the global and regional vegetation changes. It is important to discuss all regions because they are all different. We expect of course that many readers may be interested in only single regions, but since we divided this section in different parts according to the regions, the reader can choose which section is most interesting. We will try to shorten this chapter and will additionally include an overview sketch on the vegetation and climate change to further summarising the regional changes.

R: There's not really any Discussion section; i.e. the paper presents several lengthy Results sections (4-6) that mostly focus on describing results and do not put the results in context with the prior literature.

A: We agree that there is no classical Discussion section, at least we avoided to name a chapter "Discussion", but this classical format is not mandatory. The Result and Discussion chapter are merged in our manuscript, which has the advantage of discussing results directly and not just describing results and discussing them several sections later. With respect to the somehow independent topics of the different analysis (global, regional, linear vs. non-linear changes), we preferred this open structure. The results and discussion of the climatic background are put in the context of prior literature, but we will carefully look through the manuscript again to see if we have forgotten important literature.

R: The Conclusions are a good summary of the paper's findings. However, the last paragraph - L805-817 is an interesting paragraph arguing for the importance of the monsoons, but feels out of place here. One rule is that the Conclusions should introduce no new information; it should summarize what has come before. This paragraph reads as if it is breaking new ground rather than summarizing. Also, this is a place where better citation and discussion of the prior literature is absolutely critical. There is a *huge* literature on past monsoon dynamics, so e.g. L815-816 ("far greater importance... than previously assumed..." comes across as uninformed relative to prior work. As one example, see (Liu et al., 2004), who've previously argued for the effects of monsoonal dynamics on extra-tropical regions. So this paragraph could be a good focal topic for a discussion section.

A: We apologize if we have not clearly elaborated in previous chapters the result that the Holocene vegetation changes south of 60°N occurred predominantly in monsoon regions and regions that may be influenced by the changes in the monsoon dynamic. In the revised manuscript, we will stress this more clearly in chapter 5 and will add a short discussion on the effect of monsoons on the extratropical atmospheric dynamics based on previous literature. We are aware of several studies analysing the monsoon teleconnections with the extratropics. With the statement (L815-816) referred to in this comment we have not claimed that the importance of the monsoons on the extratropics has not been shown before in previous studies, but we are not aware of any study that emphasize the dominant role of the monsoon in determining the Holocene global vegetation change. This is an important result of our study. At this point, we would also like to stress that the effect of monsoons on extratropics is not the topic of the paper, but a result/explanation of the analysis on the vegetation change.

Line-by-Line Comments:
L22: Here, elsewhere, capitalize Northern Hemisphere and Southern Hemisphere

A: In different British dictionaries, we find different spelling. Hence we leave the decision of "Northern Hemisphere" vs "northern hemisphere" to the Climate of the Past copy editors.

L39: This focus on monsoons is appropriate, but I would argue that monsoons are only half the story. The other big change is the northern hemisphere tundra/treeline ecotone, which is probably more of a direct response to T and insolation, instead of monsoons. See e.g. L290-295
A: We agree, and that's why we have specified the region (...outside the high northern latitudes) in the same sentence..

L45-54: This opening intro on orbital forcing should also mention changes in obliquity/tilt at least in passing.
A: The obliquity/tilt is not a strong forcing during the Holocene, therefore we have left this out. We will change the sentence to: This contributed mainly to the gradual changes in the seasonal energy input from the sun.

L52: Here and elsewhere: spell out 'approx.' as approximately
A: Will be done in the revised version.

L56: Yes to changes in annual mean signals, but the sentence implies that only mean

annual signals were changed.  Rephrase.

A:  In the revised version, we will delete this sentence to shorten the introduction.

L56-57:  warmer/colder than what?  Be sure that all comparative statements like this have a clear referent.

A: We will add "compared to present-day" to this statement.

L61-62:  Over what time period is this weakening/strengthening?

A: We think in the context of the sentence before, it is clear that the time period is the Holocene.

L65:  Here and elsewhere, check for verb tense consistency.  This sentence is in past tense, while most of the prior paragraph was in present tense.

A: We agree and will carefully check the verb tenses.

L85-87:  This section understates the existing literature on Holocene land cover reconstructions.  Other papers:  (Williams, 2003) (Williams et al., 2011) (Pirzamanbein et al., 2018; Pirzamanbein et al., 2014) (Trondman et al., 2015) (Marquer et al., 2017).

A: These are, of course, very important studies on the vegetation reconstruction side, some of which we have cited elsewhere. The sentence was originally meant to refer to syntheses that go beyond a single continent and cover time-series. As far as we know, the study by Cao et al. 2019 covers the largest region and longest time period. We have not phrased this well. Therefore, we will add most of the proposed references.

L88-90:   This sentence understates the availability of Holocene paleoclimatic proxies, which has improved greatly over the last several years.  Papers include: (Kaufman et al., 2020) (Marsicek et al., 2018) and the authors should also do a search for 'The Holocene Conundrum' to find additional recent appers.

A: We are aware of these studies, but most proxies are still marine or based on vegetation. To specify this statement, we add 'terrestrial climate records'. We have analysed The Holocene Conundrum as seen in the MPI-ESM Holocene simulations in an earlier paper by Bader et al. (Nature Communications, 2020).

L98:  I would argue that vegetation changes are complex everywhere, not just in North America, with all areas experiencing distinct PFT-level and taxa-level changes.

A: That's the main point. The vegetation changes in North America are rather on taxa-level and only in a few regions visible on mega-biome level. Other large parts of the world experienced substantial PFT- or even mega-biome changes. To avoid confusions and to shorten the introduction, we will delete this sentence in the revised manuscript

L101-102:  Clarify that this forest expansion is during the mid to late Holocene and is in the eastern Great Plains.

A: We agree and will write: Furthermore, forest taxa have re-invaded into the prairie (eastern Great Plains), reflecting the increase in precipitation over the mid- and late Holocene (Grimm et al., 2001).

L102-103: I think this sentence needs to be narrowed to South America, not all of the Americas, but check the original reference.

A: Thank you for pointing to this, this sentence was indeed not precise enough. We meant the savanna regions on both sides around the equator. We add "equatorial savanna regions".

L120: It's not clear why this fairly lengthy review of the prior literature indicates we lack substantial understanding. I'd argue that we actually have a pretty good understanding of past vegetation changes. A better argument would be: 1) We have quite a detailed understanding of the patterns of Holocene vegetation dynamics, thanks to lots of regional to continental to hemispheric-scale syntheses. 2) We still lack a clear understanding of the climatic and other drivers (and feedbacks) associated with these vegetation changes. This shift from pattern to process would then help motivate the modeling study presented here.

A: That is exactly what we have intended to say with this sentence. We will modify it to phrase it more clearly.

L195: spin-up simulation? I haven't heard the 'spin-down' phrase before.

A: We agree that the term "spin-down" may be confusing for most readers. The reason for using this term was that the mid-Holocene equilibrium simulation was branched off from the pre-industrial control run and ran somehow 'back in time' until mid-Holocene conditions have been approached. To distinguish this method technically from the spin-up of the PI control run, we called this "spin-down". However, to avoid this confusion we change "spin-down" to "spin-up" in the revised manuscript.

L199-200: Given all these recent papers using the same model and setup, what new is being contributed by this paper?

A: The other papers deal with totally different topics, they are not about vegetation changes, with the exception that the end of the African humid period has been analysed on the basis of the simulated bare soil fraction. But this study was also based on an older simulations. For this study on the global vegetation, a simulation with improved forcing has been used.

L233: exclude->minimize (there was some land use prior to 150BC), see e.g. (Stephens et al., 2019) (Mottl et al., 2021).
A: This is correct, but in the model, land use has been prescribed only for the last ~2000 years (LUH2 dataset + ramp-up). In the model, land use therefore only affects the climate after 2.15ka. We are aware of the fact, that some reconstructions reveal strong land cover changes even earlier than this period, but this land use is "unknown" to the model. Before the year 2.15ka it simulates only the natural vegetation. As forcing data, we need global assessment of the land use, that's why we prescribed the LUH2 dataset that is also used in the CMIP6 simulations. We do not change this in the revised manuscript.

L233-239: This treatment of timescales is very confusing. Suggest ditching entirely the

BC/BCE timescale and only using ka BP, using yr 2000 (b5k) as the datum. Most paleoclimatologists and paleoecologists use ka BP, not BCE.

A: We think that using only ka BP or ka b2k is fine for the result and discussion part, but it is better to define the time-slices in the method part to be precise. The problem is that in palaeo-modelling the year 2000 is commonly used as reference, whereas palaeo-reconstructions commonly use the year 1950. So "ka BP" means ka before 1950, "ka b2k" means ka before the year 2000. This difference is not unimportant and it is advised by the Climate of the Past editors to distinguish it with these nomenclature. Therefore, we will have to keep this paragraph.

L262-264: Add some of the other Holocene vegetation references noted above. Williams et al. 2011 provides a Northern Hemisphere reconstruction for the extratropics.

A: Because we focus this sentence on biomes, we would like to refer to biome reconstruction only and not listing studies on quantitative reconstruction.

L269: Here and elsewhere: Delete sentences that solely exist to introduce a figure. Just describe a key result and put a figure pointer in parentheses at end of sentence (Fig. 1).

A: Thank you for this advise. We consider this issue as a matter of personal style. Specifically this sentence is clearly needed to introduce the figure.

L273-275: 'On one hand... on other hand...' isn't appropriate here, because the two sentences aren't really opposed to each other.

A: we will change this to 'furthermore'

L277: Not reflected in what way? Not shown at all?

A: We will change 'reflected' to 'reproduced'

L279: What is meant by a 'very distinct ecology?

A: Savannas exist in climatic zones that are suitable for forest and grasslands. In some, tree fraction is very dense, in some tree fraction is quite low. While tropical savannas require the coexistence of trees and C4-grass, they can only be distinguished from forests by their unique functional ecology, fire tolerance and shade intolerance. We further specify this in the revised manuscript.

L305-306: This description of c-means clustering is redundant with methods.
A: We agree and will shorten this introductory sentence.

L310-311: rephrase to avoid double negative
A: We will change this to: However, non-linear changes in PFTs may occur on local or regional level.

L354: At some point, these RDA analyses could be compared to remote sensing analyses of climatic drivers, e.g. (Seddon et al., 2016) (Seddon et al., 2016)
A: Thank you for this reference. There are indeed interesting similarities. We will compare our results with Seddon et al in the revised manuscript.

Fig 1 legend: 'positive ecological development' and 'negative ecological development' is too vague and sounds normative. Simply use 'increase in openness' and 'decrease in

openness'
A: We agree and we will change the label.

Fig 4b could be deleted; I'm not sure how much information it adds to what's shown in Fig. 4a.
A:We agree and we will merge the figures.

Fig 5: This figure legend is wordy. Keep the legend focused on figure orientation, and move other material as needed to the main text.
A: We will shorten the caption and move detailed explanations on the calendar problem to the main text.